# A Scalable Constant-Factor Approximation Algorithm for $W_p$ Optimal Transport

**Pankaj K. Agarwal**[1*]**, Oliver Chubet**[2]**, Sharath Raghvendra**[2]**, and Keegan Yao**[1]

[1]Duke University, [2]North Carolina State University

## Abstract

Let $(X, d)$ be a metric space and let $\mu, \nu$ be discrete probability distributions supported on finite point sets $A, B \subseteq X$. For any $p \in [1, \infty]$, the $W_p$-*distance* between $\mu$ and $\nu$, $W_p(\mu, \nu)$, is defined as the $p$-th root of the minimum cost of transporting all the probability mass from $\mu$ to $\nu$, where moving a probability mass of $\delta$ from $a \in A$ to $b \in B$ incurs a cost of $\delta \mathsf{d}(a,b)^p$. We give a (Las Vegas) randomized algorithm that computes a $(4 + \varepsilon)$-approximate $W_p$ optimal-transport (OT) plan in $O(n^2 + (n^{3/2}\varepsilon^{-1} \log n \log \Delta)^{1+o(1)} \log U)$ time with probability at least $1 - 1/n$, for all $p \in [1, \infty]$, where $\varepsilon > 0$ is an arbitrarily small constant and $\Delta$ is the ratio between the largest and smallest interpoint distances in $A \cup B$. The previous best result achieved an $O(\log n)$-approximation in $O(pn^2)$ time, for constant values of $p$. Our algorithm significantly improves the approximation factor and, importantly, is the first quadratic-time method that extends to the $W_\infty$-distance. In contrast, additive approximation methods such as Sinkhorn are efficient only for constant $p$ and fail to handle $p = \infty$. Our algorithm also extends to a query model where, for any integer $k > 1$, we give an algorithm that preprocesses $X$ into clusters in $O(n^2 + kn^{1+1/k} \log n \log \Delta)$ time, after which a $O(k)$-approximate $W_p$ distance between any two distributions $\mu$ and $\nu$ with $X$ as support can be computed in $(n^{1+1/k} \log n \log \Delta)^{1+o(1)}$ time with probability at most $1 - 1/n$. Finally, for $p = \infty$, we show that obtaining a relative approximation factor better than 2 in $O(n^2)$ time would resolve the long-standing open problem of computing a perfect matching in an arbitrary bipartite graph in quadratic time.

## 1 Introduction

Let $\mu$ and $\nu$ be discrete probability distributions supported on sets $A$ and $B$, respectively, with $|A| + |B| = n$. For each pair $(a, b) \in A \times B$, let $\mathsf{d}(a, b)$ denote the ground distance between $a$ and $b$. A *transport plan* is a function $\sigma : A \times B \to \mathbb{R}_{\geq 0}$ that assigns a mass to each pair $(a, b)$ such that $\sum_{b \in B} \sigma(a, b) \leq \mu(a)$ and $\sum_{a \in A} \sigma(a, b) \leq \nu(b)$. Given a parameter $p \geq 1$, suppose the cost of moving a probability mass of $\delta$ from a point $a \in A$ to a point $b \in B$ is given by $\mathsf{d}(a, b)^p$. The $W_p$ cost of any transport plan $\sigma$ between $\mu$ and $\nu$ is defined as

$$w_p(\sigma) := \left( \sum_{a \in A, b \in B} \sigma(a, b) \times \mathsf{d}(a, b)^p \right)^{1/p}.$$

The $W_p$ cost above for finite $p$ extends naturally to the $W_\infty$ cost of a transport plan $\sigma$, defined as

$$w_\infty(\sigma) := \lim_{p \to \infty} w_p(\sigma) = \max_{a, b \in A \times B \,:\, \sigma(a,b) > 0} \mathsf{d}(a, b).$$

In the $W_p$ *optimal transport* (OT) problem, we wish to compute the transport plan $\sigma^*$ that transports the entire mass and has the smallest $W_p$ cost. We refer to the cost $w_p(\sigma^*)$ as the $W_p$-*distance* and denote it by $W_p(\mu, \nu)$. If $\mu$ and $\nu$ are discrete distributions, each supported on $n$ points with every

---

*Following convention from Theoretical Computer Science, all authors are ordered alphabetically.

point assigned a mass of $1/n$, then the corresponding $W_p$ optimal transport problem is called the $W_p$ *matching* problem.

The $W_p$ distances, for varying values of $p$, possess several appealing properties that make them favorable in many applications. OT plans under the $W_1$ distance measure total displacement and are therefore useful to capture structural properties such as semantic relationships from word embeddings (Kusner et al. (2015)). OT plans arising from the $W_2$ distance have nice structural qualities such as monotonicity (Brenier (1991); Aurenhammer et al. (1998)) and translation invariance (Cohen & Guibas (1999)), and tend to preserve the geometry of the distributions. Furthermore, recent work in machine learning and topological data analysis uses $W_\infty$ distance to establish consistency and convergence properties of topological summaries (Vishwanath et al. (2020); Damrich et al. (2024)), and to design topological layers in neural networks (Kim et al. (2020)). Due to these favorable properties, $W_p$ distances have been used in applications across machine learning (Chang et al. (2023); Chuang et al. (2022)), computer vision (Backurs et al. (2020); **?**), and natural language processing (Alvarez-Melis & Jaakkola (2018); Yurochkin et al. (2019)).

From an algorithmic standpoint, the exact computation of the $W_p$ distance between discrete distributions can be formulated as a minimum-cost flow (MCF) problem, which can be solved in $n^{2+o(1)}$ time using recent advances in MCF algorithms (Chen et al. (2022)). While these results mark important theoretical progress, they are highly complicated, making them unsuitable for practical implementations. Indeed, even the simpler task of designing a truly quadratic-time exact algorithm for deciding whether a dense graph admits a perfect matching remains a longstanding open problem in graph theory (Behnezhad et al. (2024)).

Because practical exact algorithms remain expensive, research has shifted toward scalable approximation methods. In this setting, an algorithm is said to compute a relative $\alpha$-approximation for $W_p(\mu, \nu)$ if it outputs a transport plan $\sigma$ whose cost satisfies $w_p(\sigma) \leq \alpha \cdot W_p(\mu, \nu)$. A seminal result by Charikar (2002) introduced an $O(\log n)$-approximation for $W_1$ by embedding the ground metric into a hierarchically well-separated tree; a greedy transport procedure on the tree yielded an exact solution in $O(n^2)$ time, producing an overall $O(\log n)$-approximation. This work opened the door to more refined methods, and subsequent efforts developed near-linear-time $(1 + \varepsilon)$-approximation algorithms under additional assumptions, such as when the ground distance is Euclidean in fixed dimensions (Agarwal et al. (2022; 2024); Fox & Lu (2023)), and more recently, sub-quadratic algorithms for higher-dimensional Euclidean settings (Andoni & Zhang (2023); Beretta et al. (2025)). However, none of these techniques extend naturally to the case $p \geq 2$. Building on this line of work, Lahn et al. (2025) recently presented a relative $O(\log n)$-approximation algorithm for any finite $p \geq 2$ and for any metric, with runtime $O\left(n^2 \log U \log \Delta \log n\right)$, where $\log U$ is the bit-length of the input probabilities and $\Delta$ is the spread of $A \cup B$ (the ratio of its largest to smallest nonzero pairwise distance).

One influential direction of work was introduced by Cuturi (2013), who proposed entropic regularization of OT. It guarantees solutions within an additive error of $\varepsilon \operatorname{diam}(A \cup B)$, where $\operatorname{diam}(A \cup B)$ denotes the maximum ground distance between points in $A \cup B$. Although weaker than a relative $(1 + \varepsilon)$-approximation, it applies across all metrics and inspired a series of additive approximation algorithms, including parallelizable variants (Altschuler et al. (2017); Dvurechensky et al. (2018); Jambulapati et al. (2019); Lahn et al. (2019; 2023)). Nonetheless, their runtimes remain on the order of $n^2/\varepsilon^{O(1)}$ for $W_1$ and worsen to $n^2/\varepsilon^{O(p)}$ for larger $p$, with no extension to the case $p = \infty$.

**Our results.** In this paper, we present two constant-factor approximation algorithms for the $W_p$ problem. These are the first truly quadratic-time (assuming $\Delta + U = 2^{O(n^{1/8})}$) approximation algorithms for the $W_p$ problem over *any ground metric*, applicable to all $p \in [1, \infty]$; that is, including $p = \infty$. This improves the $O(\log n)$-approximation of Lahn et al. (2025) to a constant factor, while extending the guarantee to every $p$.

**Theorem 1.1.** *Let $\mu$ and $\nu$ be two discrete distributions supported on a set of $n$ points in an arbitrary metric space. Let $p \in [1, \infty]$ be a parameter, and let $\varepsilon > 0$ be an arbitrarily small constant. A $(4 + \varepsilon)$-approximate OT plan under the $W_p$ metric can be computed in $O(n^2 + (n^{3/2}\varepsilon^{-1} \log \Delta \log n)^{1+o(1)} \log U)$ time with probability at least $1 - 1/n$, where $\Delta$ is the spread of the support set and $U$ is the ratio of the maximum to minimum (non-zero) probability in $\mu$ or $\nu$.*

For any integer $k > 1$, we construct a directed spanner of size $O(n^{3/2} \log \Delta)$ (for $k = 2$), and more generally $O(kn^{1+1/k} \log \Delta)$ for arbitrary $k$. We prove that the shortest-path distance from any $b \in B$ to any $a \in A$ in the spanner approximates $\mathsf{d}(a, b)^p$ within a multiplicative factor of $(4 + \varepsilon)^p$ for $k = 2$, and $(2k(1 + \varepsilon))^p$ for general $k$. Running a minimum-cost flow algorithm on this directed spanner then yields an approximate optimal transport plan; in particular, instantiating it with the algorithm of Chen et al. (2022) and setting $k = 2$ implies Theorem 1.1.

Our main technical contribution is a new technique for approximating $\mathsf{d}(\cdot, \cdot)^p$, drawing on ideas from Bourgain's multi-level sampling Bourgain (1985). Multi-level sampling has inspired numerous clustering-based constructions, including metric spanners and distance oracles. Classical spanner constructions Har-Peled et al. (2023); Baswana & Sen (2007); Cohen (1998) require super-quadratic preprocessing time and approximate only the underlying metric. However, for $p > 1$, the transport cost is $\mathsf{d}(\cdot, \cdot)^p$, which is not a metric (it fails the triangle inequality), and therefore cannot be handled by these undirected metric spanners. Our spanner differs from prior constructions in two key ways. First, we introduce carefully chosen Steiner points at multiple scales of the clustering, which substantially simplify the structure of the spanner and reduce its construction time to $O(n^2 + kn^{1+1/k} \log \Delta)$ which is a significant improvement over classical spanners. Second, to approximate $\mathsf{d}(\cdot, \cdot)^p$, we assign a direction to each edge, producing a directed spanner. Although our directed spanner need not be strongly connected, it guarantees that there exists a path from any $b \in B$ to any $a \in A$. We show that it approximates the cost $\mathsf{d}(a, b)^p$ from any $b \in B$ to any $a \in A$ within a factor of $(4 + \varepsilon)^p$ (when $k = 2$) and $(2k(1 + \varepsilon))^p$ (for arbitrary $k$).

**Query-model interpretation.** Theorem 1.1 naturally extends to a *query model*, where the support set $X$ is fixed in advance and many distributions supported on $X$ must be compared. This setting arises frequently in natural language processing. For example, a document can be represented as a probability distribution over fixed word embeddings (Kusner et al. (2015)); the similarity between two documents is measured by the $W_1$ distance between these distributions, referred to as the Word Mover's Distance (WMD). Kusner et al. (2015) nevertheless note that the quadratic (or worse) running time of optimal transport is a major barrier to the scalability of WMD while processing large set of documents. In contrast, for any integer $k > 1$, our algorithm performs a one-time $O(n^2 + kn^{1+1/k} \log \Delta)$ preprocessing of $X$, after which each $W_p$ (or $W_\infty$) query between any distributions supported on $X$ can be answered by using any MCF solver on a sparse directed graph of size $O(kn^{1+1/k} \log \Delta)$ for an $O(k)$-approximation. Thus, in applications like WMD where many OT computations share the same support, our method yields a sub-quadratic query time.

Since the interior-point method of Chen et al. (2022) does not admit a practical implementation, we also develop a simple combinatorial algorithm for optimal transport. A crucial ingredient in implementing combinatorial algorithms efficiently is the ability to answer *dynamic weighted bichromatic closest-pair (BCP)* queries quickly. Existing work has used Bourgain's sampling-based clusterings to construct *distance oracles*, which return approximate point-to-point distances (Thorup & Zwick (2005); Mendel & Naor (2007); Awerbuch et al. (1998)). However, while such oracles are well suited for distance estimation, it appears extremely challenging to adapt them to support *dynamic* BCP queries, which require repeatedly identifying the minimum over all cross pairs under changing weights. To address this gap, we tailor our clustering scheme so that it naturally supports dynamic weighted BCP queries. Importantly, although pairwise-distance computations are expensive in our framework, our algorithms avoid them entirely and rely solely on the BCP primitive. Using these data structures with standard implementation of combinatorial OT algorithms (Vaidya (1989)), we obtain a simple combinatorial algorithm that computes, for any integer $k > 1$ an $O(k)$-approximation to the $W_p$ optimal-transport cost in $O(n^{2+1/k} \log \Delta \log U \log^2 n)$ time (Theorem 1.2), as well as a quadratic time $(4 + \varepsilon)$-approximate matching algorithm (Theorem 1.3), complementing the quadratic OT algorithm obtained through the interior-point method of Chen et al. (2022) (Theorem 1.1).

**Theorem 1.2.** *Let $(X, d)$ be an arbitrary metric space, let $\mu$ and $\nu$ be two probability measures supported on finite point sets $A, B \subseteq X$ with $|A \cup B| = n$, and let $p \in [1, \infty]$ be a parameter. For any integer $k > 1$, an $O(k)$-approximation to the $W_p$ optimal-transport cost $W_p(\mu, \nu)$ can be computed in $O(n^{2+1/k} \log^2 n \log \Delta \log U)$ time with probability at least $1 - 1/n$, where $\Delta$ is the spread of $A \cup B$ and $\log U$ is the maximum to minimum (non zero) probability at $\mu$ and $\nu$.*

**Theorem 1.3.** *Let $A$ and $B$ be two point sets of size $n$ each in an arbitrary metric space, and let $p \in [1, \infty]$ be a parameter. A $(4 + \varepsilon)$-approximate $W_p$ matching of $A$ and $B$ can be computed*

in $O(n^2\varepsilon^{-3/2}\log^2 n\log\Delta)$ *time with probability at least* $1-1/n$, *and an $O(k)$-approximate $W_p$ matching of $A$ and $B$ can be computed in $O(n^2+n^{3/2+1/k}\varepsilon^{-3/2}\log^2 n\log\Delta)$ time with probability at least* $1-1/n$.

To our knowledge, Theorems 1.2 and 1.3 provide the first practical and implementable approximation algorithm for computing the $W_\infty$-OT in sub-$n^{2.5}$ time and $W_\infty$-matching in quadratic time. We also establish conditional lower bounds that suggest our results cannot be significantly improved without a major breakthrough in the graph matching problem, namely, computing a perfect matching in any bipartite graph in $O(n^2)$ time.

**Theorem 1.4.** *If there exists a quadratic-time algorithm that computes a $(2-\varepsilon)$-relative approximation or $\mathrm{diam}/2-\varepsilon$ additive approximation of the $W_\infty$-matching problem, where $\mathrm{diam}$ is the diameter of the point set and $\varepsilon>0$ is any arbitrarily small constant, then a perfect matching in an arbitrary bipartite graph can be computed in $O(n^2)$ time if one exists.*

We conclude with a primitive implementation of our simple combinatorial algorithm alongside some experimental results suggesting that the algorithm computes good quality $W_p$-matchings for $p\in[1,\infty]$ in Section 4. While we prove that the approximation factor of our algorithm is $(4+\varepsilon)$ in the worst case, our experimental results indicate that our algorithm computes even better approximations of $W_p$-matchings in practice.

## 2 DISTANCE APPROXIMATION AND PROXIMITY QUERIES

Let $P$ be a set of points equipped with a metric $\mathsf{d}\colon P\times P\to\mathbb{R}_{\geq 0}$ and let $k>1$ be an integer. We introduce a $k$ level clustering-based distance function that approximates $\mathsf{d}(\cdot,\cdot)$ within a multiplicative factor of $2k(1+\varepsilon)$. Our construction is similar in spirit to the construction of $k$-spanners and distance oracles discussed in Section 1. Our distance can be represented using roughly $kn^{1+1/k}$ space (as opposed to $O(n^2)$ space to store all pairwise distances), and is used both to build a spanner and to maintain weighted bichromatic closest pairs. For simplicity in presentation, we describe our clustering construction and associated algorithms for $k=2$ in the paper. The extension to any arbitrary integer $k>2$ is deferred to the Appendix B.

We begin with a few notations. Given a point $x\in P$ and a subset $Q\subseteq P$, the distance from $x$ to $Q$ is defined as $\mathsf{d}(x,Q)=\min_{q\in Q}\mathsf{d}(x,q)$. For a point $q\in P$ and subset $Q\subseteq P$, define the *Voronoi set* of $q$ to be

$$V(q,Q):=\{y\in P\mid \mathsf{d}(y,q)<\mathsf{d}(y,Q)\}.$$

That is, $V(q,Q)$ consists of the points in $P$ for which $q$ is closer than any point in $Q$.

**Two-level clustering.** We construct a two-level clustering of points of $P$. Set $P_0=P$. Next, we choose a subset $P_1\subseteq P_0$ by sampling each point in $P_0$ independently with probability $n^{-1/2}$. The expected size of $P_1$ is $\mathbb{E}[|P_1|]=n^{1/2}$.

Let $\Delta$ be the spread of the point set $P$. Without loss of generality, assume $\min_{p,q\in P}\mathsf{d}(p,q)=1$ implying that the diameter $\max_{p,q\in P}\mathsf{d}(p,q)$ is $\Delta$. We choose $\varepsilon>0$ to be a sufficiently small constant. Set $t=\lceil\log_{(1+\frac{\varepsilon}{4})}\Delta\rceil$, $r_0=0$, and $r_i=(1+\frac{\varepsilon}{4})^i$ for $1\leq i\leq t$. We generate two types of clusters: (i) For each $q\in P_0\setminus P_1$ and for every $i\leq t$, define $C_q[i]=\{x\in V(q,P_1)\mid \mathsf{d}(x,q)\leq r_i\}$. (ii) For each $q\in P_1$ and for every $i\leq t$, define $C_q[i]=\{x\in P_0\mid \mathsf{d}(x,q)\leq r_i\}$. We refer to $i$ as the *index* of the cluster $C_q[i]$. Let $\mathcal{C}=\{C_q[i]\mid q\in P_0, i\leq t\}$ be the collection of all clusters. Note that a point $p\in P$ may belong to many clusters. The number of clusters that contain $p$ is called the *degree* of $p$ and is denoted as $\deg_{\mathcal{C}}(p)$. While the degree of any particular point may be $\Theta(n)$ in the worst case, we prove that, with high probability, the degree of all points in $P$ is much smaller.

**Lemma 2.1.** *For all $p\in P$, $\mathbb{E}[\deg_{\mathcal{C}}(p)]=O(\sqrt{n}\varepsilon^{-1}\log\Delta)$. Furthermore, there is a constant $c>0$ such that $\Pr[\forall p\in P,\deg_{\mathcal{C}}(p)\leq c\sqrt{n}\varepsilon^{-1}\log n\log\Delta]\geq 1-1/n$.*

*Proof.* We first bound the number of clusters of type (ii). Each point of $P_0$ is selected independently with probability $1/\sqrt{n}$, so $\mathbb{E}[|P_1|]=\sqrt{n}$. By a Chernoff bound, $|P_1|\leq 2\sqrt{n}$ with probability at least $1-\frac{1}{n^2}$. Each point $q\in P_1$ generates $O(\varepsilon^{-1}\log\Delta)$ clusters of type (ii), and any point $p$ may participate in each such cluster. Therefore, the expected number of type (ii) clusters that $p$

participates in is $O(\sqrt{n}\,\varepsilon^{-1}\log\Delta)$, and with probability at least $1 - 1/n^2$, the number of type (ii) clusters that $p$ participates in is bounded by $O(\sqrt{n}\,\varepsilon^{-1}\log\Delta)$.

We now bound the contribution from clusters of type (i). For each $q \in P_0 \setminus P_1$, we compute its Voronoi set $V(q, P_1)$ and then construct $O(\varepsilon^{-1}\log\Delta)$ clusters from that region. It therefore suffices to bound the number of Voronoi sets that contain a fixed point $p \in P$. Fix $p \in P$. Let $w_1, \ldots, w_{n-1}$ be the points of $P_0$ ordered in non-decreasing distance from $p$. Let $s$ be the smallest index such that $w_s \in P_1$.

For any index $\ell > s$, we have $\mathsf{d}(p, w_\ell) > \mathsf{d}(p, w_s)$. Since $w_s \in P_1$, the point $p$ is closer to $w_s$ than to $w_\ell$, and hence $p \notin V(w_\ell, P_1)$. Thus, $p$ can belong to $V(w_\ell, P_1)$ only if $\ell < s$, and therefore the number of Voronoi sets containing $p$ is at most $s$. Since each point is selected into $P_1$ independently with probability $1/\sqrt{n}$, the random variable $s$ is geometric with success probability $1/\sqrt{n}$. Hence, the expected value of $s$ is $\mathbb{E}[s] = \sqrt{n}$. Moreover,

$$\Pr[s > t] = \left(1 - \frac{1}{\sqrt{n}}\right)^t \le e^{-t/\sqrt{n}}.$$

Choosing $t = 2\sqrt{n}\log n$, we get $\Pr[s > 2\sqrt{n}\log n] \le e^{-2\log n} = n^{-2}$. Therefore, the number of type (i) clusters containing $p$ is $O(s\,\varepsilon^{-1}\log\Delta)$, which is $O(\sqrt{n}\,\varepsilon^{-1}\log\Delta)$ in expectation and $O(\sqrt{n}\log n\,\varepsilon^{-1}\log\Delta)$ with probability at least $1 - 1/n^2$. Combining the bounds for type (i) and type (ii) clusters and applying a union bound completes the proof. $\square$

Our algorithms and their running-time bounds assume that every vertex $q \in P$ satisfies $\deg_C(q) = O(\sqrt{n}\,\varepsilon^{-1}\log n\log\Delta)$. Since this bound holds with probability at least $1 - 1/n$, the stated execution-time guarantees also hold with the same probability. Also, because each point participates in $O(\sqrt{n}\,\varepsilon^{-1}\log n\log\Delta)$ clusters, the total space required to explicitly store all clusters is $O(n^{3/2}\,\varepsilon^{-1}\log\Delta)$ in expectation and $O(n^{3/2}\,\varepsilon^{-1}\log n\log\Delta)$ with probability at least $1 - 1/n$.

**Constructing clusters.** Constructing clusters of type (i) proceeds as follows. For each point $q \in P_0 \setminus P_1$, we first compute its closest point in $P_1$ by scanning $P_1$. The Voronoi set $V(q, P_1)$ is then determined by checking, for every $q' \in P_0$, whether $q'$ is closer to $q$ than to its nearest neighbor in $P_1$. This requires $O(n)$ time per $q$. Once $V(q, P_1)$ is obtained, the associated clusters are formed by partitioning its points according to their distance from $q$, which can be done with a single additional scan. Thus, clusters of type (i) can be constructed in $O(n)$ time per $q \in P_0 \setminus P_1$, and $O(n^2)$ time overall. For clusters of type (ii), each point $q \in P_1$ induces a partition of $P_0$ based on distance from $q$, which takes $O(n)$ per $q$ and $O(n^2)$ time in total.

**Cluster-induced distance approximation.** Using this two-level clustering, we now define a distance function $\mathsf{d}_C: P \times P \to \mathbb{R}_{\ge 0}$ that approximates the ground metric within a factor of $(4 + \varepsilon)$. For any pair of points $x, y \in P$, let $i$ be the smallest index of a cluster that contains both $x$ and $y$. Then we set $\mathsf{d}_C(x, y) = 2r_i$.

**Lemma 2.2.** $\mathsf{d}(x, y) \le \mathsf{d}_C(x, y) \le (4 + \varepsilon)\mathsf{d}(x, y)$.

*Proof.* We say a point $x$ is separated from $y$ by $P_1$ if there is a point $a \in P_1$ such that $\mathsf{d}(x, a) < \mathsf{d}(x, y)$. From this and triangle inequality, $\mathsf{d}(y, a) \le \mathsf{d}(x, a) + \mathsf{d}(x, y) \le 2\mathsf{d}(x, y)$. Let $i$ be the smallest index such that $y \in C_a[i]$. Then, $\mathsf{d}(y, a) \le r_i < (1 + \frac{\varepsilon}{4})\mathsf{d}(y, a)$. So we have $\mathsf{d}_C(x, y) \le 2r_i \le 2\left(1 + \frac{\varepsilon}{4}\right)\mathsf{d}(y, a) \le 4\left(1 + \frac{\varepsilon}{4}\right)\mathsf{d}(x, y)$. If $x$ is not separated from $y$, then $x \in V(y, P_1)$. Let $i$ be the smallest index such that $C_y[i]$ contains $x$. Then, $\mathsf{d}_C(x, y) \le 2r_i \le 2\left(1 + \frac{\varepsilon}{4}\right)\mathsf{d}(x, y)$. $\square$

### 2.1 Proximity Queries

Next, we show how to use the above clustering to construct a directed spanner and dynamic data structures that support proximity queries with respect to $\mathsf{d}_C(\cdot, \cdot)$. Using these structures, in Section 3, we obtain efficient approximation algorithms.

**Directed spanner.** Let $A, B \subseteq P$ be two disjoint subsets of $P$, let $\mathsf{d}: P \times P \to \mathbb{R}_{\ge 0}$ be a metric, and let $p \in [1, \infty)$. We construct a graph $G = (V, E)$ and a set of edge weights $w_p: \overline{E} \to \mathbb{R}_{\ge 0}$ such that the shortest path from any $a \in A$ to any $b \in B$ in the graph $G$ with respect to weights $w_p$ is approximately $\mathsf{d}^p(a, b)$.

For each cluster $C \in \mathcal{C}$, we create two vertices $a_C, b_C$. Set $V = A \cup B \cup \{a_C, b_C \mid C \in \mathcal{C}\}$. For each cluster $C \in \mathcal{C}$, we add the following three sets of edges to $E$:

(i) Add the edge $a_C \to b_C$ and set $w_p(a_C \to b_C) = (2r_i)^p$ if the index of $C$ is $i$.

(ii) For every $a \in A \cap C$, we add the edge $a \to a_C$ and set $w_p(a \to a_C) = 0$.

(iii) For every $b \in B \cap C$, we add the edge $b_C \to b$ and set $w_p(b_C \to b) = 0$.

Clearly $|V| = O(n\varepsilon^{-1} \log \Delta)$ since $|\mathcal{C}| = O(n\varepsilon^{-1} \log \Delta)$. Since the degree of each point is $O(\sqrt{n}\varepsilon^{-1} \log n \log \Delta)$, the number of edges is $O(n^{3/2}\varepsilon^{-1} \log n \log \Delta)$. Define $\mathsf{d}_{G,p} \colon A \times B \to \mathbb{R}_{\geq 0}$ as the shortest path distance in $G$ with respect to edge weights $w_p$.

**Lemma 2.3.** *The weighted graph $G$ with weights $w_p$ satisfies $\mathsf{d}^p(a,b) \leq \mathsf{d}_{G,p}(a,b) \leq (4+\varepsilon)^p \cdot \mathsf{d}^p(a,b)$ for all $a,b \in A \times B$ and for any $p \in [1,\infty)$.*

**Dynamic bichromatic closest pair.** Let $P$ be a point set, $p \geq 1$ be an integer, and let $A, B \subseteq P$ be two disjoint subsets. Given a weight function $w \colon A \cup B \to \mathbb{R}$, we define the *weighted distance* $\mathsf{d}_w \colon A \times B \to \mathbb{R}$ as

$$\mathsf{d}_w(a,b) = \mathsf{d}_{\mathcal{C}}^p(a,b) - w(a) - w(b).$$

Our goal is to maintain $\mathrm{BCP}_w(A,B) = \arg\min_{(a,b) \in A \times B} \mathsf{d}_w(a,b)$ under insertions and deletions of points in $A$ and $B$, where all inserted points come from $P$. We now describe a simple data structure that supports maintaining $\mathrm{BCP}_w(A,B)$ dynamically.

We construct the clustering $\mathcal{C}$ over the entire set $P$. For each cluster $C \in \mathcal{C}$, we maintain the points in $B \cap C$ in a max-heap keyed by their weights, and the points in $A \cap C$ in a max-heap keyed by their weights. Let $a^C$ (resp., $b^C$) denote the point at the root of the heap for $A \cap C$ (resp., $B \cap C$). If $C$ has index $i$, we define $\phi_C = (2r_i)^p - w(a^C) - w(b^C)$. We then maintain the set $\mathsf{X} = \{(a^C, b^C) \mid C \in \mathcal{C}\}$ in a global min-heap $H$, using $\phi_C$ as the key. The pair at the root of $H$ is precisely $\mathrm{BCP}_w(A,B)$. The following observation is critical to the design of the BCP data structure.

**Lemma 2.4.** *Let $(a^*, b^*)$ be the pair stored at the root of $H$. Then $\mathsf{d}_w(a^*, b^*) = \min_{a,b \in A \times B} \mathsf{d}_w(a,b)$.*

A similar claim also appears in Lahn et al. (2025) as Lemma 2.3. We include the proof in Appendix C, for completeness. Inserting or deleting a point $q \in A$ (resp., $q \in B$) requires visiting all clusters containing $q$ and updating the corresponding max-heap. If the root of any such heap changes, we update the global heap accordingly. Since each point participates in $\deg_{\mathcal{C}}(q) = O(\sqrt{n}\varepsilon^{-1} \log n \log \Delta)$ clusters, the total update time is $O(\sqrt{n}\varepsilon^{-1} \log^2 n \log \Delta)$.

**Lemma 2.5.** *Let $P$ be a set of $n$ points in a metric space and let $\mathcal{C}$ be its two-level clustering. Let $A, B \subseteq P$ be two weighted point sets. There is a data structure that maintains the BCP under the distance function $\mathsf{d}_w$ in $O(1)$ time while allowing for insertion or deletion of points, where inserting or deleting a point $q \in P$ takes $O(\deg_{\mathcal{C}}(q) \log n)$ time.*

# 3 ALGORITHMS FOR $W_p$

In this section, we use the collection of clusters and data structures constructed in Section 2 to design two efficient algorithms for the optimal transport problem.

## 3.1 MINIMUM-COST FLOW BASED ALGORITHM

Let $\mu$ and $\nu$ be discrete distributions with support sets $A$ and $B$; let $|A| + |B| = n$. We compute an approximate $W_p$-OT as follows. First assume $p \geq 1$ is a finite value. Let $G = (V, E)$ be the directed graph constructed on $A \cup B$ described in Section 2.1, and let $w_p$ be the corresponding weight function on $E$. We add a source vertex $s$ and sink vertex $t$ to the graph $G$. We also add an edge $s \to a$ for every $a \in A$ with weight $w_p(s \to a) = 0$ and an edge $b \to t$ for every $b \in B$ with weight $w_p(b \to t) = 0$. This addition gives the graph a single source and single sink to run minimum cost flow. Next, we assign capacities to each edge as follows: For each cluster $C \in \mathcal{C}$ and corresponding edge $a_C \to b_C$ in $E$, assign a capacity of $u(a_C \to b_C) = 1$. Additionally, for each

$a, b \in C$ we assign the capacity $u(a \to a_C) = u(b_C \to b) = 1$. Finally, for each $a \in A$ and $b \in B$, we assign the source and sink edge capacities as $u(s \to a) = \mu(a)$ and $u(b \to t) = \nu(b)$.

We compute the capacitated min-cost max- flow $f^*$ in this directed graph using the algorithm by Chen et al. (2022) in $(|E|)^{1+o(1)} \log U$ time. Using the minimum cost flow $f^*$, we compute a transport plan $\sigma$ where $w_p(\sigma) \leq \left( \sum_{e \in E} f^*(e) w_p(e) \right)^{1/p}$ as follows. Initially, $\sigma(a, b) = 0$ for all $a \in A, b \in B$. While the total flow from $s$ to $t$ is positive, find any path $\pi = s \to a \to a_C \to b_C \to b \to t$ where $f^*(e) > 0$ for every edge $e$ on the path $\pi$ and increment $\sigma(a, b)$ by $\lambda = \min\{f^*(e) \mid e \in \pi\}$. Additionally decrement $f^*(e)$ by $\lambda$ for every edge $e \in \pi$. We repeat until $f^*$ is zero everywhere. This concludes the construction of the transport plan $\sigma$. It follows naturally from Lemma 2.3 that $\sigma$ is an approximate transport plan with respect to the $W_p$ distance. Since $|E| = O(n^{3/2} \varepsilon^{-1} \log n \log \Delta)$ edges with probability at least $1 - 1/n$, the overall runtime of constructing $\sigma$ from $f^*$ is $O(n^{3/2} \varepsilon^{-1} \log n \log \Delta)$. This proves Theorem 1.1 for $p \in [1, \infty)$.

For $p = \infty$, we proceed as follows. We maintain the same source and sink vertices $s, t$ as well as the same edge capacities as above. We then compute a sequence of maximum flows instead of a single minimum cost flow in $G$, and perform binary search on the radii of the clusters. By construction, there are at most $O(\varepsilon^{-1} \log \Delta)$ different values of $r_i$. For a fixed $1 \leq i \leq O(\varepsilon^{-1} \log \Delta)$, define the graph $G_i$ to be the graph $G$ with all edges of cost $w_1(a_C \to b_c) > 2r_i$ removed. Compute a maximum flow $f_i$ in $G_i$ from $s$ to $t$ in $(n^{3/2} \varepsilon^{-1} \log n \log \Delta)^{1+o(1)} \log U$ time using the algorithm of Chen et al. (2022). If $\sum_{a \in A} f_i(s \to a) = 1$, then conclude that $W_\infty(\mu, \nu) \leq 2r_i$ and decrease $i$. Otherwise, conclude that $W_\infty(\mu, \nu) > 2r_i$ and increase $i$.

Let $i^*$ be the smallest value of $i$ such that $\sum_{a \in A} f_i(s \to x) = 1$. Then we compute a transport plan $\sigma$ from $f_{i^*}$ as above in the case when $p < \infty$. Initially, $\sigma(a, b) = 0$ for all $a \in A, b \in B$. While the total flow from $s$ to $t$ is positive, find any path $\pi = s \to a \to a_C \to b_C \to b \to t$ where $f_{i^*}(e) > 0$ for every edge $e$ on the path $\pi$ and increment $\sigma(a, b)$ by $\lambda = \min\{f_{i^*}(e) \mid e \in \pi\}$. Additionally decrement $f_{i^*}(e)$ by $\lambda$ for every edge $e \in \pi$. We repeat until $f_{i^*}$ is zero everywhere. This concludes the construction of the transport plan $\sigma$. It follows naturally from Lemma 2.3 that $\sigma$ is an approximate transport plan with respect to the $W_\infty$ distance. Similar to the algorithm for finite $p$, we observe that the overall runtime of constructing $\sigma$ from $f_{i^*}$ is $O(n^{3/2} \varepsilon^{-1} \log n \log \Delta)$. This proves Theorem 1.1 for $p = \infty$.

## 3.2 COMBINATORIAL ALGORITHMS

Classical capacity-scaling algorithms for the $W_p$-OT perform $O(n \log U)$ Hungarian searches (i.e., dual-adjustment steps) to compute a solution. Each Hungarian search can be implemented by inserting/deleting each point of $A \cup B$ (resp., $A$) into a BCP data structure at most once Lahn et al. (2025); Atkinson & Vaidya (1995); Agarwal & Sharathkumar (2012). Since inserting or deleting a point $q$ requires time proportional to $O(\deg_{\mathcal{C}}(q) \log n)$, a single Hungarian search can be executed in $O(\log n \sum_{q \in A \cup B} \deg_{\mathcal{C}}(q))$ time, which is $O(kn^{1+1/k} \log^2 n)$ for any integer $k > 1$. Theorem 1.2 follows from the fact that the algorithm performs $O(n \log U)$ such Hungarian searches. For the $W_p$-matching problem, the number of Hungarian searches can be reduced to $O(\sqrt{n})$ Agarwal & Sharathkumar (2012; 2014), yielding the overall bound stated in Theorem 1.3. In this section, we describe the resulting combinatorial algorithm for $k = 2$, which runs in $\tilde{O}(n^2)$ time and computes a $(4 + \varepsilon)$-approximate $W_p$-matching between two point sets $A$ and $B$ of size $n$.

We begin by constructing the two-level clustering $\mathcal{C}$ on the input point set $A \cup B$. We describe the algorithm and analyze its running time under the assumption that the degree of every points of $A \cup B$ is $O(\sqrt{n}, \varepsilon^{-1} \log n \log \Delta)$. Since this degree bound holds with probability at least $1 - 1/n$, the same high-probability guarantee applies to the overall algorithm. The algorithm selects an appropriate scaling parameter $\delta > 0$ and simulates a single scale of the Gabow-Tarjan cost-scaling algorithm for bipartite matching. We begin by defining scaled costs as a scaled version of the $p$-th power of the proxy distance $\hat{c}(a, b) = \left\lceil \frac{1}{\delta} d_{\mathcal{C}}^p(a, b) \right\rceil$. The algorithm proceeds with these integer costs $\hat{c}(a, b)$.

**Matchings and augmenting paths.** A *matching* $M$ is a collection of vertex–disjoint edges. A vertex not incident to any edge of $M$ is said to be *free*. A matching is *perfect* if no vertex is free. Given a matching $M$, an *alternating path* is a path whose edges alternate between those in $M$ and those

outside $M$. An *augmenting path* is an alternating path whose two endpoints are free. Augmenting along such a path flips the membership of its edges in $M$, thereby increasing the size of $M$ by one.

**1-feasible matching.** Each vertex $v \in A \cup B$ is assigned an integer dual variable $y(v)$. A matching $M$ and dual weights $y(\cdot)$ are *1-feasible* if

$$
\begin{aligned}
y(a) + y(b) &\leq \hat{c}(a,b) + 1 && \text{for all } (a,b) \in A \times B, && (1) \\
y(a) + y(b) &= \hat{c}(a,b) && \text{for all } (a,b) \in M. && (2)
\end{aligned}
$$

We define the *slack* of an edge $(a,b)$ with respect to a matching $M$ and dual weights $y(\cdot)$ as

$$
s(a,b) = \begin{cases} 0, & \text{if } (a,b) \in M, \\ \hat{c}(a,b) - y(a) - y(b) + 1, & \text{if } (a,b) \notin M. \end{cases}
$$

An edge is *admissible* if $s(a,b) = 0$, and the set of admissible edges forms the *admissible graph*

We initialize the matching $M = \emptyset$ and set all dual weights to zero, i.e., $y(v) = 0$ for every $v \in A \cup B$. Note that $(M, y)$ is 1-feasible. Let $B_F = B$ be the free vertices of $B$ with respect to $M$. The algorithm maintains a 1-feasible pair $(M, y)$ consisting of a matching $M$ and dual weights $y(\cdot)$, and executes iterations. Each iteration has the *dual adjustment* step, which builds an augmenting path of admissible edges, and *augmentation* step, which computes a maximal set of vertex-disjoint augmenting paths and augments the matching along these paths to increase the size of the matching. Next, we describe the dual adjustment and the augmentation steps.

**Dual adjustment via BCP-based Hungarian Search.** The Hungarian search procedure runs a Dijkstra-style shortest path search using slacks as edge lengths. This search is implemented using bichromatic closest pair (BCP) queries, described in Section 2.1, with implicit dual updates. The search maintains a tree. Let $U \subseteq B$: the set of vertices of $B$ already added to this search tree and let $V \subseteq A$ be the set of vertices of $A$ not yet added to the search tree. Initially, $U$ contains all free vertices in $B$, each with distance label $\ell_b = 0$, and $V = A$. Define effective weights

$$
w(b) = \delta(y(b) - \ell_b) \quad \text{for } b \in U, \qquad w(a) = \delta\, y(a) \quad \text{for } a \in V.
$$

At each iteration, select the edge $(a,b) = \arg\min_{a' \in V, b' \in U}\{s(a',b') + \ell_{b'}\}$ which we show (in Appendix D) is exactly the weighted BCP maintained by our data structure, i.e., $\arg\min_{a' \in V, b' \in U}\{\mathrm{d}_{\mathcal{C}}^p(a',b') - w(a') - w(b')\}$. Remove $a$ from $V$, set $\ell_a = \ell_b + s(a,b)$, and add it to the search tree. If $a$ is free, an augmenting path has been found and the search terminates. Otherwise, let $b'$ be its matched partner; set $\ell_{b'} = \ell_a$, update $w(b') = y(b') - \ell_{b'}$, and insert $b'$ into $U$. Once the search terminates, the dual weights are updated as

$$
y(a) \leftarrow y(a) - \ell_{a^\star} + \ell_a \quad \text{for all } a \in S, \qquad y(b) \leftarrow y(b) + \ell_{a^\star} - \ell_b \quad \text{for all } b \in T,
$$

where $S \subseteq A$ and $T \subseteq B$ are the sets of vertices reached.

*Efficiency of Hungarian Search.* The BCP data structure maintains the weighted closest pair between the sets $U$ and $V$. Initially, we set $U = B_F$ and $V = A$. In each iteration, one point of $A$ is removed from $V$, and at most one point of $B \setminus U$ is inserted into $U$. Inserting a point $u$ into the data structure requires $O(\deg_{\mathcal{C}}(u) \log n)$ time. Since each vertex in $A \cup B$ is inserted or deleted at most once, the total running time of a Hungarian search is $O\left(\sum_{u \in A \cup B} \deg_{\mathcal{C}}(u) \log n\right) = O(n^{3/2} \varepsilon^{-1} \log \Delta \log^2 n)$.

**Augmentation step.** Once the dual-adjustment phase reaches a free vertex in $A$, the search guarantees the existence of at least one augmenting path in the admissible graph. The algorithm then computes a maximal set of vertex-disjoint augmenting paths via a sequence of partial depth-first searches (DFS). We initiate a DFS from each free vertex of $B$, one at a time. Let $X \subseteq A$ denote the set of vertices in $A$ that have not yet been visited by any DFS; initially, $X = A$. For every cluster $C \in \mathcal{C}$, we maintain a max-heap over the points in $X \cap C$, keyed by $y(a)$ for $a \in X$. In addition, each point $q \in A \cup B$ maintains the list of clusters in which it participates.

The DFS alternates between unmatched admissible edges from $B$ to $A$ and matched edges from $A$ back to $B$. When the search is at a vertex $u \in B$, it scans the clusters containing $u$. For each such cluster $C$ (say of index $i$), let $v$ be the root of the max-heap at $C$. It checks if $\lceil (2r_i)^p / \delta \rceil - y(u) -$

$y(v)+1 = 0$, i.e., $(u, v)$ is admissible. If not, we remove $C$ from the cluster list of $u$ and continue to the next cluster. If an admissible edge to some vertex $v$ is found, we remove $v$ from $X$ (i.e., delete $v$ from all max-heaps corresponding to clusters that contain $v$) and extend the alternating path. If $v$ is matched to some $u'$, we append the edge $(v, u')$ to the path and continue the DFS from $u'$. If instead $v$ is free, an augmenting path has been discovered. We prove the correctness of the DFS procedure in Lemma D.2. After completing a DFS, the algorithm proceeds with the next free vertex of $B$. The process terminates once a DFS has been initiated from every free vertex in $B$. Finally, all discovered augmenting paths are flipped simultaneously to update the matching. For each augmenting path and for every vertex $b \in B$ lying on it, reduce the dual weight by one: $y(b) \leftarrow y(b)-1$. This correction guarantees that all newly matched edges remain tight under the 1-feasible condition.

The algorithm executes dual adjustment and augmentation until all vertices are matched.

*Efficiency of partial DFS.* Initializing the max-heaps across all clusters requires $O(n^{3/2}\varepsilon^{-1} \log \Delta \log^2 n)$ time. When exploring a vertex $u$, we scan the clusters in its cluster list. If a cluster does not yield an admissible edge, it is removed from the list. Each cluster can be removed from $u$'s list at most once, so the total number of such events for a vertex $u$ is bounded by $\deg_{\mathcal{C}}(u)$ for $u$ and cumulative across all $u \in A$, the time taken by these events can be bounded by $O(n^{3/2}\varepsilon^{-1} \log \Delta \log^2 n)$. Finally, when an admissible edge is found at a vertex $u$, $u$ is deleted from $X$ requiring max-heap updates which take $O(\deg_{\mathcal{C}}(u) \log n)$ time. Since each vertex is processed at most once in this manner, the overall cost remains at $O(n^{3/2}\varepsilon^{-1} \log \Delta \log^2 n)$. Thus, the total running time of the partial DFS phase is $O(n^{3/2}\varepsilon^{-1} \log \Delta \log^2 n)$.

**Overall Efficiency.** We select the parameter $\delta$ (using $O(\log n)$ guesses) so that the edge costs become integers and the sum of the costs of the edges in the optimal matching (referred to as *total cost*) is scaled to $\Theta(n/\varepsilon)$. Scaling by $\delta$ preserves the true optimum total cost, while rounding introduces at most an additive error of $n$ in the total cost. Moreover, the 1-feasible matching produced is itself within $+n$ of the rounded optimum (Gabow & Tarjan (1989)). Hence, the total deviation is at most $2n$, and whenever the rounded optimum is at least $2n/\varepsilon$, the resulting solution is guaranteed to be within a $(1 + \varepsilon)$ factor of the true optimum. Gabow and Tarjan showed that if the costs are integers and the value of the optimal solution is $O(n/\varepsilon)$, then a single scale of their algorithm converges in $O(\sqrt{n/\varepsilon})$ phases. In particular, when the optimal total cost is $\leq 2n/\varepsilon$, the algorithm terminates in $O(\sqrt{n/\varepsilon})$ phases, and combined with the error bounds above, produces a $(1 + \varepsilon)$-approximation. Each of the two steps—dual adjustment and augmentation—of a phase can be implemented in $O(n^{3/2}\varepsilon^{-1} \log \Delta \log^2 n)$. Combined across $O(\sqrt{n/\varepsilon})$ phases, the total execution time is bound in Theorem 1.3.

## 4 EMPIRICAL EVALUATION

This section contains an empirical evaluation of the clustering method from Section 2 as well as the approximation factor obtained from the primal–dual algorithm of Section 3.2. We also compare the performance of our algorithm to that of Lahn et al. (2025). Both methods compute an optimal matching with respect to a proxy distance using the same primal–dual framework. Our approach employs a clustering-based proxy distance (*cluster-dist*) that approximates true distances within a worst-case factor of $(4 + \varepsilon)$, whereas Lahn et al. (2025) uses an HST-based proxy distance (*HST-dist*) with an $O(\log n)$ distortion guarantee. Computations were performed on a computer with an 8-core Apple M1 CPU with 16GB RAM. Samples are drawn from uniform and truncated normal distributions on the unit cube in up to 10 dimensions, and we additionally evaluate the algorithm on samples of the MNIST dataset LeCun (1998), treated as 784-dimensional data.

**Proxy distance accuracy.** We first evaluate the quality of the clustering by comparing the induced cluster distances to the ground metric. For each value of $n$, we measure both the maximum and the average distortion across all pairs. Figure 1e confirms that the worst-case distortion never exceeds the theoretical $(4+\varepsilon)$-approximation guarantee of Lemma 2.2, and that the average distortion is often substantially smaller—typically close to a factor of 2—indicating that the effective approximation in practice is far tighter than the worst-case bound. For comparison, we perform the same evaluation for the HST-dist of Lahn et al. (2025) for $p = 2$ and on the uniform distribution. As shown in Figure 1e, both the average and the maximum distortion incurred by HST-dist are considerably larger than those of cluster-dist, highlighting the improved accuracy of cluster-dist.

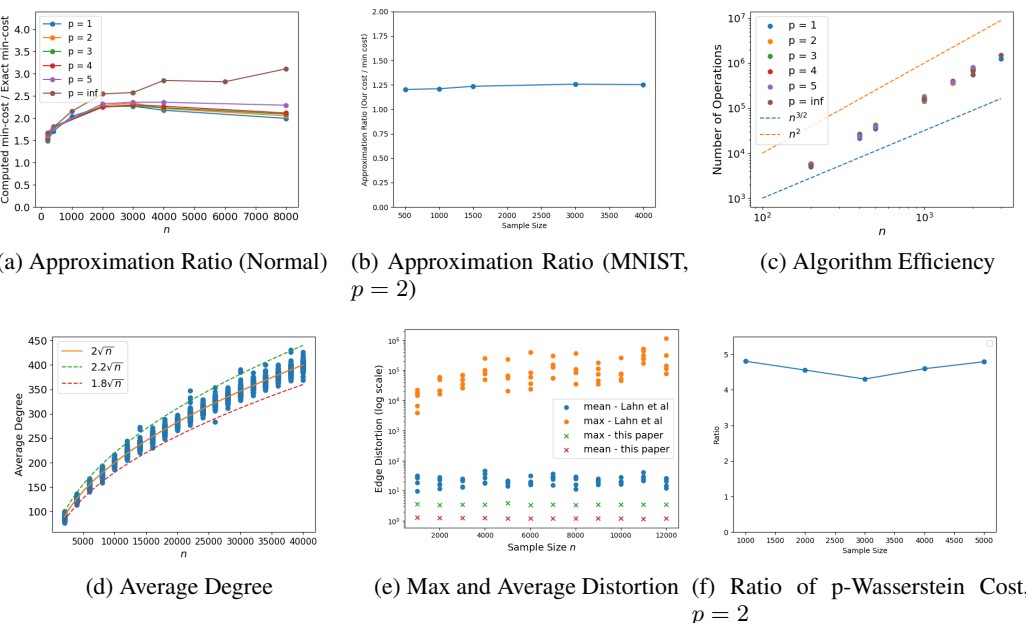

(a) Approximation Ratio (Normal)     (b) Approximation Ratio (MNIST, $p = 2$)     (c) Algorithm Efficiency

(d) Average Degree     (e) Max and Average Distortion     (f) Ratio of p-Wasserstein Cost, $p = 2$

Figure 1: Empirical evaluation of the 2-layer clustering and $W_p$-matching algorithm.

**Algorithm accuracy.** To evaluate the accuracy of the primal–dual matching algorithm, we compare the computed matching cost to that obtained using the exact distance matrix. Figures (2) and (1a) report the approximation ratio across values of $p \in \{1, 2, 3, 4, 5, \infty\}$. The ratios consistently remain well within the theoretical $(4+\varepsilon)$ factor, with typical values close to $1.5$–$2$, again suggesting that the empirical performance is considerably better than the worst-case analysis. This trend is stable across both uniform and normal distributions, as well as MNIST data. We further provide the ratio of the optimal matching costs under the HST-dist and Cluster-dist for $p = 2$ and the uniform distribution in Figure 1f. The results demonstrate that the optimal matching cost computed using cluster-dist is consistently more than $4.5$ times smaller than the cost obtained using HST-dist.

**Clustering efficiency.** Next, we examine the degree, ie. the number of clusters each point participates in. Figure (1d) shows that the observed average degrees closely track the theoretical bound of Lemma 2.1 for dimension $d \le 10$, and both distributions. This indicates that the two-layer clustering is both space-efficient and stable across different settings.

**Algorithm efficiency.** We measure efficiency by the number of bichromatic closest pair (BCP) queries, which dominate the running time. As shown in Figure (1c), the query counts scale as predicted and remain nearly identical across all choices of $p$. Combined with the $\tilde{O}(n^{1/2})$ per-query complexity, this provides strong empirical evidence that the algorithm runs in quadratic time and scales smoothly with problem size.

**Summary.** Overall, the experiments demonstrate that the proposed method is both theoretically grounded and empirically robust. While the theoretical analysis guarantees only a $(4 + \varepsilon)$ approximation, the observed approximation ratios are consistently much smaller, indicating that the algorithm performs substantially better in practice. The clustering step is efficient in both time and space, and its distortions are far below the worst-case bound. These results suggest that our approach is a practical alternative to additive methods such as Sinkhorn, particularly in regimes where existing techniques either fail to apply (e.g., $p = \infty$) or require higher-than-quadratic time.

## 5 REPRODUCIBILITY

All experimental results were produced using a proof-of-concept python implementation. The source code for the implementation can be found at `https://anonymous.4open.science/r/anon_matching-2B3B/`.

## ACKNOWLEDGEMENT

Work by Pankaj Agarwal and Keegan Yao has been partially supported by NSF grants CCF-22-23870 and IIS-2402823 , and by US-Israel Binational Science Foundation Grant 2022/131. Work by Oliver Chubet and Sharath Raghvendra is supported by an NSF Grant CCF-2514753. We would like to thank the anonymous reviewers for their useful feedback.

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

# A    CONDITIONAL HARDNESS OF THE $W_\infty$-MATCHING PROBLEM

In this section, we prove a conditional hardness result for approximating the $W_\infty$ distance.

Suppose we are given the bipartite graph $G = (V, E)$, where $V = V_1 \cup V_2$ with $|V_1|, |V_2| = n$, $V_1 \cap V_2 = \varnothing$, and $E \subseteq V_1 \times V_2$. We will reduce this problem of determining if $G$ contains a perfect matching to computing an approximate $W_\infty$-distance.

Construct the following metric $\rho$ on $V$. For each distinct $v_1, v_2 \in V$, if $(v_1, v_2) \in E$ or $(v_2, v_1) \in E$ then define $\rho(v_1, v_2) = 1$. Otherwise, define $\rho(v_1, v_2) = 2$. For completeness, one can define $\rho(v, v) = 0$ for all $v \in V$. The finite metric space $(V, \rho)$ can be constructed in $O(n^2)$ time, given $G$. It is easy to see that $(V, \rho)$ is a metric space: (i) by definition, $\rho(v, v) = 0$ and $\rho(v, v') > 0$ for all $v' \neq v$; (ii) if $\rho(v_1, v_2) = 1$ for some $v_1 \neq v_2$, then either $(v_1, v_2)$ or $(v_2, v_1)$ is in $E$, in which case $\rho(v_2, v_1) = 1$; (iii) $\rho(v_1, v_2) \leq 2$ and $\rho(v_1, v_3) + \rho(v_3, v_2) \geq 1 + 1 = 2$ for all $v_1, v_2 \in V$ and $v_3 \neq v_1, v_2$, implying triangle inequality in combination with observation (i) to prove the degenerate case $v_3 = v_1$ or $v_3 = v_2$.

Additionally define the distributions $\mu_1 \colon V_1 \to [0, 1]$ and $\mu_2 \colon V_2 \to [0, 1]$ by $\mu_1(v_1) = \mu_2(v_2) = \frac{1}{n}$ for all $v_1 \in V_1$ and all $v_2 \in V_2$. Assume we are given an approximation algorithm $\mathcal{A}$ for the $W_\infty$ distance, and let $\sigma_{\mathcal{A}}$ denote the transport plan from $\mu_1$ to $\mu_2$ computed by algorithm $\mathcal{A}$.

Since $\rho(v_1, v_2) \in \{1, 2\}$ for all $v_1, v_2 \in V_1 \times V_2$, it must also be the case that

$$w_\infty(\sigma) := \max_{x, y : \sigma(x, y) > 0} \rho(x, y) \in \{1, 2\}$$

for any transport plan $\sigma$ and therefore $W_\infty(\mu_1, \mu_2) \in \{1, 2\}$. We use this observation to prove the following crucial relationship between $W_\infty(\mu_1, \mu_2)$ and the maximum cardinality matching in $G$.

**Lemma A.1.** *A perfect matching exists in $G$ if and only if $W_\infty(\mu_1, \mu_2) = 1$.*

*Proof.* If $W_\infty(\mu_1, \mu_2) = \min_\sigma w_p(\sigma) = 2$, then it is impossible to construct a transport plan $\sigma$ where every edge has distance 1 and therefore at least one edge $(u, v)$ where $\sigma(u, v) > 0$ satisfies $\rho(u, v) = 2$. By definition, the metric $\rho$ is equal to 2 if and only if the pair is not an edge in $G$. Therefore, any matching must have size at most $n - 1$.

If $W_\infty(\mu_1, \mu_2) = 1$, then it is possible to construct a transport plan $\sigma$ where every edge has a distance 1, and by standard network flow theory this transport plan $\sigma$ is a convex combination of matchings in $(V_1 \cup V_2, V_1 \times V_2)$. By definition of $\rho$, we note $\rho(v_1, v_2) = 1$ if and only if $(v_1, v_2) \in E$. We conclude that any matching $M$ where $\sigma(v_1, v_2) > 0$ for every $(v_1, v_2) \in M$ is also a perfect matching in $G$. $\qquad\square$

Therefore, it suffices to determine if either $W_\infty(\mu_1, \mu_2) = 1$ or $W_\infty(\mu_1, \mu_2) = 2$, and extract any matching from the resulting transport plan in $O(n^2)$ time in the former case.

We briefly describe this (standard) procedure of how to construct a perfect matching $M$ in $V_1 \times V_2$ using $\sigma_{\mathcal{A}}$ in $O(n^2)$ time such that for every $(v_1, v_2) \in M$, we have $\sigma_{\mathcal{A}}(v_1, v_2) > 0$. To do this, we iteratively choose an arbitrary edge $(u, v) \in V_1 \times V_2$ such that $\sigma(u, v) > 0$, add $(u, v)$ to $M$, remove $u$ from $V_1$ and $v$ from $V_2$. Repeat until $M$ is a perfect matching. Since $\sigma$ is known to be a convex combination of perfect matchings, we conclude that the algorithm always constructs a perfect matching.

What follows is an observation that even when given relatively weak approximation algorithms for the $\infty$-Wasserstein distance, it is possible to distinguish between these two cases. We first prove this for the case that $\mathcal{A}$ is a relative approximation algorithm.

**Lemma A.2.** *Suppose for some $\varepsilon > 0$ there exists an algorithm running in $O(n^2)$ time which computes a $(2 - \varepsilon)$-approximate transport plan under the $W_\infty$ metric between two discrete probability distributions with supports of size at most $n$. Then, for any bipartite graph $G$, one can compute a perfect matching in $G$ or conclude that none exists in $O(n^2)$ time.*

*Proof.* We assume the existence of an algorithm $\mathcal{A}$ which, when given an input finite metric space $(V, \rho)$ and distributions $\mu, \nu$, computes a transport plan $\sigma$ such that $w_\infty(\sigma) \leq (2 - \varepsilon) \cdot W_\infty(\mu, \nu)$ in $O(n^2)$ time.

First suppose that $w_\infty(\sigma_\mathcal{A}) = 2$. By the approximation guarantee of the algorithm $\mathcal{A}$, we conclude that $W_\infty(\mu_1, \mu_2) \geq \frac{2}{2-\varepsilon} > 1$ and therefore $W_\infty(\mu_1, \mu_2) = 2$. Now suppose that $w_\infty(\sigma_\mathcal{A}) = 1$. Then we have immediately found a minimizing transport plan and conclude that $W_\infty(\mu_1, \mu_2) = 1$. We conclude that if $\mathcal{A}$ is a $(2 - \varepsilon)$-approximation algorithm then necessarily $w_\infty(\sigma_\mathcal{A}) = W_\infty(\mu_1, \mu_2)$ for the constructed input metric space $(V, \rho)$ and distributions $\mu_1, \mu_2$. The result follows after a simple application of Lemma A.1. $\qquad\square$

We emphasize that the metric space used to prove Lemma A.2 has spread $\Delta = 2$. Therefore, Lemma A.2 is useful even if the algorithm $\mathcal{A}$ has $O(n^2 f(\Delta))$ runtime for any function $f \colon \mathbb{R}_{\geq 0} \to \mathbb{R}_{\geq 0}$ independent of $n$. Next, we prove the analogous claim for the case where $\mathcal{A}$ is an additive approximation algorithm.

**Lemma A.3.** *Suppose for some $\varepsilon > 0$ there exists an algorithm running in $O(n^2)$ time which computes a $(\frac{\Delta}{2} - \varepsilon)$-additive approximate transport plan under the $W_\infty$ metric between two discrete probability distributions with supports of size at most $n$ in a metric space with diameter $\Delta$. Then, for any bipartite graph $G$, one can compute a perfect matching in $G$ or conclude that none exists in $O(n^2)$ time.*

*Proof.* We assume the existence of an algorithm $\mathcal{A}$ which, when given an input finite metric space $(V, \rho)$ and distributions $\mu, \nu$, computes a transport plan $\sigma$ such that $w_\infty(\sigma) \leq W_\infty(\mu, \nu) + \left(\frac{\Delta}{2} - \varepsilon\right)$ in $O(n^2)$ time.

First suppose that $w_\infty(\sigma_\mathcal{A}) = 2$. By the approximation guarantee of the algorithm $\mathcal{A}$, we conclude that $W_\infty(\mu_1, \mu_2) \geq w_\infty(\sigma_\mathcal{A}) - \left(\frac{\Delta}{2} - \varepsilon\right) = 2 - (1 - \varepsilon) > 1$ and therefore $W_\infty(\mu_1, \mu_2) = 2$. Now suppose that $w_\infty(\sigma_\mathcal{A}) = 1$. Then we have immediately found a minimizing transport plan and conclude that $W_\infty(\mu_1, \mu_2) = 1$. We conclude that if $\mathcal{A}$ is a $\left(\frac{\Delta}{2} - \varepsilon\right)$-approximation algorithm then necessarily $w_\infty(\sigma_\mathcal{A}) = W_\infty(\mu_1, \mu_2)$ for the constructed input metric space $(V, \rho)$ and distributions $\mu_1, \mu_2$. The result follows after a simple application of Lemma A.1. $\qquad\square$

Then a simple combination of Lemmas A.2 and A.3 implies Theorem 1.4.

# B $\quad k$-LEVEL CLUSTERING

In Section 2, we used a two-layered clustering to approximate d up to multiplicative factor $4 + \varepsilon$ using $O(n^{3/2})$ space. This approach can be generalized to a $k$-level clustering. Extending to $k$-level clustering has the benefit of a reduced expected degree of each point in $P_0$, at the expense of an increase in the guaranteed stretch factor of the data structure.

In this section, we describe a clustering based distance function that can be constructed in $O(n^2 + kn^{1+1/k} \log \Delta)$ time, approximates d up to a factor of $2k(1 + \varepsilon)$ and that can be represented using $O(kn^{1+1/k} \varepsilon^{-1} \log \Delta)$ space. Similar to Section 2.1, we can use this clustering to construct a spanner and maintain (weighted) bichromatic closest pairs. Then once we have those data structures, the algorithms in Section 3 work in the same manner. Applying the combinatorial algorithm in Section 3.2 to the $k$-level clustering for $k = 3$ gives the $(6 + \varepsilon)$-approximation result in Theorem 1.3.

$k$-**level clustering.** We now construct a $k$-level clustering of points of $P$. Let $P_0 = P$. For $i = 1, \ldots, k - 1$, we next choose a subset $P_i \subseteq P_{i-1}$ by sampling each point in $P_{i-1}$ independently with probability $n^{-1/k}$. The expected size of $P_i$ is $\mathbb{E}[|P_i|] = n^{1-i/k}$ for each $i \leq k - 1$.

Set $t = \lceil \log_{(1+\varepsilon)} \Delta \rceil$. Let $r_0 = 0$ and $r_i = (1 + \varepsilon)^i$ for $1 \leq i \leq t$. For each $0 \leq l < k - 1$, $q \in P_l \setminus P_{l+1}$ and $1 \leq i \leq t$, define the cluster $C_q[i]$ as

$$C_q[i] = \{x \in V(q, P_{l+1}) \mid \mathsf{d}(x, q) \leq r_i\}.$$

Finally for each $q \in P_{k-1}$ and $1 \leq i \leq t$, define the cluster $C_q[i]$ as

$$C_q[i] = \{x \in P_0 \mid \mathsf{d}(x, q) \leq r_i\}.$$

**Lemma B.1.** *The $k$-level clustering can be constructed in $O(n^2 + kn^{1+1/k} \varepsilon^{-1} \log \Delta)$ time.*

*Proof.* The samples $\emptyset = P_k \subset P_{k-1} \subseteq \ldots P_1 \subseteq P_0 = P$ can be computed in $O(kn)$ time.

For each $i$ from $k - 1$ to $0$, we do the following:

- Compute the distance of each $x \in P$ to the set $P_{i+1}$, where the distance from any $x \in P$ to $P_k$ is defined to be $\infty$. This takes $n|P_{i+1}|$ time, which is $O(n^{1+1/k})$ in expectation for $0 \leq i \leq k - 2$, and constant for $i = k - 1$.

- For each $p \in P_i \setminus P_{i+1}$, and for each $x \in P$ compute the Voronoi region $V(x, P_i)$ by adding $x$ to $V(x, P_{i+1})$ if $\mathsf{d}(x, p) < \mathsf{d}(x, P_{i+1})$. This takes $O(n|P_i|)$ time; that is, $O(n^{1+1/k})$ for $i > 0$ and $O(n^2)$ for $i = 0$.

- For each $p \in P_i \setminus P_{i+1}$, each $x \in V(p, P_{i+1})$, and for each index $1 \leq j \leq t$, add $C_p[j]$ if $\mathsf{d}(x, p) \leq r_j$. Throughout the entire construction, this takes time proportional to the sum of the degrees over $p \in P$, $O(kn^{1+1/k}\varepsilon^{-1} \log \Delta)$.

Thus, the clustering is computed in $O(n^2 + kn^{1+1/k}\varepsilon^{-1} \log \Delta)$ time. $\qquad\square$

Let $\mathcal{C} = \{C_q[i] \mid q \in P, i \leq t\}$ be the collection of all clusters. Again define the *degree* of a point $p \in P$ as $\deg_{\mathcal{C}}(p) := |\{C \in \mathcal{C} \mid p \in C\}|$. In an analogous manner to Lemma 2.1, we prove that the expected degree of each point in $P$ is small.

**Lemma B.2.** $\mathbb{E}[\deg_{\mathcal{C}}(p)] = O(kn^{1/k}\varepsilon^{-1} \log \Delta)$ *for all* $p \in P$.

*Proof.* We note that for any $0 \leq j \leq t$ and for any $q \in P_i \setminus P_{i+1}$ for $i < k - 1$, we have $C_q[i] \subseteq V(q, P_{i+1})$. There are at most $O(\varepsilon^{-1} \log \Delta)$ different values of $i$. Therefore it suffices to prove for any $p \in P$, the number of points $q \in P_i \setminus P_{i+1}$ where $p \in V(q, P_{i+1})$ is $O(kn^{1/k})$ in expectation.

Let $x \in P_0$ and fix $i < k - 1$. Let $w_1, \ldots, w_{n_i}$ be the elements of $P_i$ in order of non-decreasing distances to $x$. Note that $P_{i+1} \subseteq P_i$. Let $s$ be the smallest index such that $w_s \in P_{i+1}$.

For any index $r > s$ we have $\mathsf{d}(x, w_r) > \mathsf{d}(x, w_s)$. Since $w_s \in P_{i+1}$, the point $x$ is closer to $w_s$ than to $w_r$, and hence $x \notin V(w_r, P_{i+1})$. Thus $x$ can only belong to $V(w_r, P_{i+1})$ if $r < s$ and therefore the number of Voronoi sets containing $x$ at level $i$ is at most $s$. Since each point in $P_{i+1}$ is selected from $P_i$ independently with probability $n^{-1/k}$, the random variable $s$ is geometric with success probability $n^{-1/k}$. Hence, the expected value of $s_i$ is $\mathbb{E}[s] = n^{1/k}$.

$\qquad\square$

**Cluster-induced distance approximation.** As in Section 2, for any cluster $C = C_q[i] \in \mathcal{C}$ and any pair of points $x, y \in C_q[i]$, define the *cluster-induced distance* between $x$ and $y$ as

$$\mathsf{d}_C(x, y) = 2r_i.$$

Then define $\mathcal{C}(x, y) = \{C \in \mathcal{C} \mid x, y \in C\}$ and $\mathsf{d}_{\mathcal{C}}(x, y) = \min_{C \in \mathcal{C}(x,y)} \mathsf{d}_C(x, y)$. We prove that this minimum cluster-induced distance approximates $\mathsf{d}(x, y)$ within a factor or $2k(1 + \varepsilon)$.

**Lemma B.3.** $\mathsf{d}(x, y) \leq \mathsf{d}_{\mathcal{C}}(x, y) \leq 2k(1 + \varepsilon) \cdot \mathsf{d}(x, y)$

*Proof.* First we show that if $x$ and $y$ are separated by $P_l$, i.e. if there exist $\alpha, \beta \in P_l$ such that $\mathsf{d}(x, \alpha) < \mathsf{d}(x, y)$ and $\mathsf{d}(y, \beta) < \mathsf{d}(x, y)$, then $\mathsf{d}(y, P_l) \geq \mathsf{d}(y, P_{l+1}) - \mathsf{d}(x, y)$. Let $\alpha$ and $\beta$ be the closest points to $x$ and $y$ in $P_{l+1}$, respectively. Let $a$ and $b$ be the closest points to $x$ and $y$ in $P_l$ respectively. Then

$$\begin{aligned} \mathsf{d}(x, a) &\geq \mathsf{d}(y, a) - \mathsf{d}(x, y) &&\text{[triangle inequality]} \\ &\geq \mathsf{d}(y, \beta) - \mathsf{d}(x, y) &&\text{[definition of } \beta] \end{aligned}$$

Let $x$ be inserted in round $l$. Then $y \notin C_x[t]$ if and only if $\mathsf{d}(y, P_{l+1}) \leq \mathsf{d}(x, y)$. Let $C_z[t]$ be the cluster containing both $x$ and $y$ that minimizes $\mathsf{d}(y, z)$. Assume that $C_z[t]$ was inserted in round $j$. Note that $j < l$ and $l - j \leq k - 2$. Then by the claim above,

$$\mathsf{d}(y, z) \leq \mathsf{d}(y, P_{l+1}) + (l - j)\mathsf{d}(x, y)$$

$$\leq ((k - 2) + 1)\mathsf{d}(x, y).$$

Then by the triangle inequality it follows that $\mathsf{d}(x, z) \leq k\mathsf{d}(x, y)$. Note that $\mathsf{d}(x, z) \leq r_i < (1 + \varepsilon)\mathsf{d}(x, z)$, where $C_z$ has minimum index $i$ containing both $x$ and $y$. Without loss of generality assume $\mathsf{d}(x, z) \geq \mathsf{d}(y, z)$. Therefore $\mathsf{d}_{\mathcal{C}}(x, y) = 2r_i \leq 2(1 + \varepsilon)\mathsf{d}(x, z) \leq 2k(1 + \varepsilon)\mathsf{d}(x, y)$. $\qquad\square$

## C  OMITTED PROOFS FROM SECTION 2

We omitted many of the proofs of technical lemmas from the main text for the sake of space. In this section, we conclude with any missing details.

We first prove that the shortest path distance $\mathsf{d}_{G,p}$ in the directed graph $G$ with weights $w_p$ approximates $\mathsf{d}(\cdot, \cdot)^p$.

*Proof of Lemma 2.3.* Let $p \in [1, \infty)$ be an arbitrarily chosen value, and let $a \in A, b \in B$ be arbitrary points. We note by construction of $G$, for any path $\pi$ from $a$ to $b$ in the graph $G$, there exists a unique edge $e = (a_C \to b_C) \in E$ for some cluster $C \in \mathcal{C}$ such that $w(e) > 0$.

To prove the upper bound, we show that there exists a path where the edge $e = (a_C \to b_C)$ satisfies $w_p(e) \leq (4 + \varepsilon)^p \cdot \mathsf{d}^p(a, b)$. By Lemma 2.2, observe that there exists a cluster $C \in \mathcal{C}$ with index $i$ where $a, b \in C$ and $2r_i \leq (4 + \varepsilon) \cdot \mathsf{d}(a, b)$. Since $a \in C$, we observe that the edge $a \to a_C$ exists in $G$. Similarly, since $b \in C$, the edge $b_C \to b$ exists in $G$. Therefore, the path $\pi = a \to a_C \to b_C \to b$ is a path from $a$ to $b$ in $G$. The weight of the edge $a_C \to b_C$ is $w_p(a_C \to b_C) = (2r_i)^p \leq ((4 + \varepsilon) \cdot \mathsf{d}(a, b))^p = (4 + \varepsilon)^p \cdot \mathsf{d}^p(a, b)$.

To prove the lower bound, we show that for any such path $\pi$ from $a$ to $b$ in $G$, $w_p(e) \geq \mathsf{d}^p(a, b)$. Suppose $\pi = a \to a_C \to b_C \to b$ is a path from $a$ to $b$ in $G$. By construction, observe that if the path $\pi = a \to a_C \to b_C \to b$ is a path in $G$ for some cluster $C \in \mathcal{C}$ with index $i$, then both $a$ and $b$ are contained in $C$. Since $a$ and $b$ are contained in $C$, there exists a point $q \in C$ (in particular, choose the center of the cluster $C_q[i]$) such that $\mathsf{d}(a, q) \leq r_i$ and $\mathsf{d}(b, q) \leq r_i$. Conclude that $\mathsf{d}(a, b) \leq \mathsf{d}(a, q) + \mathsf{d}(q, b) \leq 2r_i$ by triangle inequality and therefore $\mathsf{d}^p(a, b) \leq (2r_i)^p = w_p(a_C \to b_C)$. $\quad\square$

We next prove that the global heap $H$ constructed for the BCP data structure will contain the weighted bichromatic closest pair at its root.

*Proof of Lemma 2.4.* For any pair $(a, b) \in A \times B$, let $C(a, b)$ denote its *witnessing cluster*, i.e., the cluster of smallest index that contains both $a$ and $b$. Let $i(C(a, b))$ denote this smallest index. By definition of $\mathsf{d}_{\mathcal{C}}$,

$$\mathsf{d}_{\mathcal{C}}(a, b) = 2r_{i(C(a,b))}.$$

Hence,

$$\mathsf{d}_w(a, b) = (2r_{i(C(a,b))})^p - w(a) - w(b).$$

Now fix a cluster $C \in \mathcal{C}$ with index $i$. Let

$$a^C \in \arg \max_{a \in A \cap C} w(a), \qquad b^C \in \arg \max_{b \in B \cap C} w(b).$$

For any $(a, b) \in (A \cap C) \times (B \cap C)$, we have

$$(2r_i)^p - w(a) - w(b) \geq (2r_i)^p - w(a^C) - w(b^C) = \phi_C.$$

Thus, among all pairs whose witnessing cluster is $C$, the smallest weighted distance is exactly $\phi_C$, achieved by $(a^C, b^C)$. Since every pair $(a, b)$ has a witnessing cluster $C(a, b)$, the inequality above implies

$$\mathsf{d}_w(a, b) \geq \phi_{C(a,b)}.$$

Taking the minimum over all pairs $(a, b) \in A \times B$ yields

$$\min_{(a,b) \in A \times B} \mathsf{d}_w(a, b) \geq \min_{C \in \mathcal{C}} \phi_C.$$

On the other hand, for every cluster $C \in \mathcal{C}$, the pair $(a^C, b^C)$ belongs to $A \times B$ and satisfies $\mathsf{d}_w(a^C, b^C) = \phi_C$. Hence,

$$\min_{(a,b) \in A \times B} \mathsf{d}_w(a, b) \leq \min_{C \in \mathcal{C}} \phi_C.$$

Combining the two inequalities gives

$$\min_{(a,b) \in A \times B} \mathsf{d}_w(a, b) = \min_{C \in \mathcal{C}} \phi_C.$$

The global heap stores one candidate per cluster with key $\phi_C$ and returns the smallest such value. Therefore, the pair at the root of the heap achieves the smallest weighted distance, as claimed. $\qquad\square$

Finally, we prove that updates to the dynamic BCP data structure can be done in $O(\sqrt{n}\varepsilon^{-1} \log n \log \Delta)$ time.

*Proof of Lemma 2.5.* Without loss of generality assume $q \in A$. There are two steps to insert a point $p$ into the $\mathrm{BCP}_w$ data structure. Similar analysis also extends to deletion as well as any point $q \in B$. For each cluster $C \in \mathcal{C}$ that contains $q$, we add $q$ to the max-heap of $A \cap C$. If the root of this heap is updated, from the global min-heap, we delete the previous pair and instead insert the new $(a^C, b^C)$ with a key $\phi_C = (2r_i)^p - w(a^C) - w(b^C)$.

Thus, the total work done is $O(\deg_{\mathcal{C}}(q) \log n)$. By Lemma 2.1, the degree of $q$ can be bounded by $O(n^{1/2} \log n \varepsilon^{-1} \log \Delta)$, leading to a insertion/deletion time of $O(n^{1/2} \log^2 n \varepsilon^{-1} \log \Delta)$ $\qquad\square$

We briefly note that the details of correctness and time complexity for the weighted nearest neighbor data structure in Section 2.1 were also omitted. The analysis of the described data structure is nearly identical to that of the dynamic BCP data structure: nearest neighbors are stored at the root of the heap, where the weight is most extreme, and insertion or deletion will modify every cluster containing the point.

## D    OMITTED PROOFS FROM SECTION 3

**Lemma D.1.** *If*

$$(a, b) = \arg \min_{a' \in V,\, b' \in U} \left\{ \mathsf{d}_{\mathcal{C}}^p(a', b') - w(a') - w(b') \right\},$$

*then* $(a, b)$ *also minimizes*

$$\arg \min_{a' \in V,\, b' \in U} \left\{ s(a', b') + \ell_{b'} \right\}.$$

*Proof.* Substituting $w(a') = \delta\, y(a')$ and $w(b') = \delta\, (y(b') - \ell_{b'})$, we obtain

$$\mathsf{d}_{\mathcal{C}}^p(a', b') - w(a') - w(b') = \mathsf{d}_{\mathcal{C}}^p(a', b') - \delta y(a') - \delta y(b') + \delta \ell_{b'}.$$

Adding $\delta$, dividing by $\delta$, and taking the ceiling does not change the minimizer. Hence $(a, b)$ also minimizes

$$\left\lceil \frac{\mathsf{d}_{\mathcal{C}}^p(a', b') - \delta y(a') - \delta y(b') + \delta \ell_{b'} + \delta}{\delta} \right\rceil.$$

Since $y(a')$, $y(b')$, and $\ell_{b'}$ are integers, and by the definition of $\hat{c}(a', b')$, this simplifies to $\hat{c}(a', b') - y(a') - y(b') + \ell_{b'} + 1$. Using $s(a', b') = \hat{c}(a', b') - y(a') - y(b') + 1$, we conclude that $(a, b)$ also minimizes $s(a', b') + \ell_{b'}$. $\qquad\square$

**Lemma D.2.** *Let $X \subseteq A$ be the set of vertices not yet visited by the DFS. Let $u \in B$ be a current leaf in the DFS tree. Let $C$ be a cluster of index $i$. If $v \in C \cap X$ is the max of the heap of $C$ keyed by dual weights and $\lceil (2r_i)^p / \delta \rceil - y(u) - y(v) + 1 = 0$, then $(u, v)$ is an admissible edge.*

*Furthermore, suppose there is any admissible edge $(u, v')$ incident to $u$ in cluster $C$. Let $v$ be the vertex of maximum dual weight. Then the edge $(u, v)$ must also be admissible.*

*Proof.* Let $X \subseteq A$ be the set of vertices not yet visited by the DFS. The first edge found in the DFS step must be a non-matching edge because the search begins at a free point of $B$. Let $(u, v)$ be an edge found in the DFS step, where $u \in B$ and $v \in X$. The edge $(u, v)$ must be a non-matching edge because whenever an edge $(b, a) \in B \times X$ is added to the the partial DFS tree, if $a$ is matched to a point $b'$, we also add $(a, b')$ and remove $a$ from $X$.

So the slack of $(u, v)$ is

$$s(u, v) = \hat{c}(u, v) - y(u) - y(v) + 1 = \lceil d_{\mathcal{C}}^p(u, v)/\delta \rceil - y(u) - y(v) + 1.$$

Let $C$ be the cluster of index $i$ from which $(u, v)$ was obtained. The algorithm additionally guarantees that

$$\lceil (2r_i)^p/\delta \rceil - y(u) - y(v) + 1 = 0.$$

To check admissibility of $(u, v)$ we need to show that $\lceil d_{\mathcal{C}}^p(u, v)/\delta \rceil - y(u) - y(v) + 1 = 0$. So it is sufficient to show that $\lceil d_{\mathcal{C}}^p(u, v)/\delta \rceil = \lceil (2r_i)^p/\delta \rceil$. By the definition of $d_{\mathcal{C}}$ we have that

$$d_{\mathcal{C}}^p(u, v) = \min_{u,v \in C[j]} (2r_j)^p.$$

So it must be true that $\lceil d_{\mathcal{C}}^p(u, v)/\delta \rceil = \lceil \min_{u,v \in C[j]} (2r_j)^p/\delta \rceil \leq \lceil (2r_i)^p/\delta \rceil$. Then by 1-feasibility we have

$$y(u) + y(v) - 1 \leq \hat{c}(u, v) \leq \lceil d_{\mathcal{C}}^p(u, v)/\delta \rceil \leq \lceil (2r_i)^p/\delta \rceil = y(u) + y(v) - 1.$$

So we have $\lceil d_{\mathcal{C}}^p(u, v)/\delta \rceil = \lceil (2r_i)^p/\delta \rceil$, implying that $(u, v)$ is admissible.

Now let $(u, v')$ be any admissible edge incident to $u$ in a cluster $C$ of index $i$. Let $v$ be the vertex of maximum dual weight. We show that the edge $(u, v)$ must also be admissible.

By the admissibility of $(u, v')$ we have $\lceil (2r_i)^p/\delta \rceil - y(u) + 1 = y(v')$. Then by 1-feasibility we have $\lceil (2r_i)^p/\delta \rceil - y(u) + 1 \geq y(v)$. Since $y(v)$ is the maximum dual weight, this implies that $y(v') = y(v)$. So $(u, v)$ is also admissible. □

## E   PROOF OF THEOREM 1.1

We omitted a formal cost analysis for the transport plan computed by the minimum-cost flow based algorithm from the main text since intuitively the fact that shortest paths in $G$ approximate $d^p$ should imply a minimum cost flow will also approximate $W_p$. In this section, we give a formal cost analysis of the algorithm in Section 3.1 for completeness. We first prove the case when $p$ is finite.

*Proof of Theorem 1.1 for $p < \infty$.* By standard analysis of network flow algorithms, since the capacity of each edge $s \to x$ is $\mu(x)$ and the capacity of each edge $y \to t$ is $\nu(y)$, we conclude that the resulting $\sigma$ is a transport plan. Moreover, given $f^*$, the transport plan $\sigma$ can be constructed in $O(|E|)$ time since every $s - t$ path in $G$ has constant length and at every iteration the flow along at least one edge is decremented to zero. We conclude with a cost analysis of $\sigma$.

First, we prove the lower bound. By construction of the graph $G$, we note that any path from $s$ to $t$ must be of the form $s \to a \rightsquigarrow b \to t$ where $a \rightsquigarrow b$ is a path in $G$ from $a \in A$ to $b \in B$. By Lemma 2.3, conclude that $\sum_{e \in a \rightsquigarrow b} w_p(e) \geq d_{G,p}(a, b) \geq d^p(a, b)$ for any $a \in A, b \in B$ and any path $a \rightsquigarrow b$ in $G$. Moreover, every edge of the form $s \to a$ and $b \to t$ has cost zero. This implies

$$W_p(\mu, \nu) \leq w_p(\sigma) = \left( \sum_{a,b \in A \times B} \sigma(a, b) \cdot d^p(a, b) \right)^{1/p} \leq \left( \sum_{e \in E} f^*(e) \cdot w_p(e) \right)^{1/p}.$$

We now prove the upper bound. Suppose we are given the optimal transport plan $\sigma^*$. We construct a flow $f$ with cost at most $(4 + \varepsilon) \cdot w_p(\sigma^*)$. For each $x, y \in A \times B$, by Lemma 2.3 there exists a path $x \rightsquigarrow y$ in $G$ such that $\sum_{e \in x \rightsquigarrow y} w_p(e) \leq (4 + \varepsilon) \cdot d(x, y)$. Let $\pi(x, y)$ denote this specific path for each pair $x, y$.

Initially, set $f$ to be zero everywhere. Then for each pair $x, y$ where $\sigma^*(x, y) > 0$, we increment flow $f$ on every edge along $s \to x, y \to t$ and every edge of $\pi(x, y)$ by $\sigma^*(x, y)$. The resulting flow

is a max flow since the capacity on every edge $s \to x$ leaving $s$ is $\mu(x)$ and the capacity on every edge $y \to t$ entering $t$ is $\nu(y)$. Moreover by construction of $f$ and the fact that every edge $s \to x$ and $y \to t$ has cost zero, the cost of the flow $f$ is

$$\left( \sum_{e \in E} f(e) \cdot w_p(e) \right)^{1/p} = \left( \sum_{x,y:\sigma^*(x,y)>0} \sigma^*(x,y) \cdot \mathsf{d}_{G,p}(x,y) \right)^{1/p}$$

$$\leq \left( \sum_{x,y:\sigma^*(x,y)>0} \sigma^*(x,y) \cdot ((4+\varepsilon)^p \cdot \mathsf{d}^p(x,y)) \right)^{1/p}$$

$$\leq (4+\varepsilon) \cdot w_p(\sigma^*) = (4+\varepsilon) \cdot W_p(\mu,\nu).$$

We note that the minimum cost flow must have a cost no more than $f$. □

Then the cost analysis for when $p = \infty$ follows in a similar manner.

*Proof of Theorem 1.1 for $p = \infty$.* By standard analysis of network flow algorithms, since the capacity of each edge $s \to x$ is $\mu(x)$ and the capacity of each edge $y \to t$ is $\nu(y)$, we conclude that the resulting $\sigma$ is a transport plan. Moreover, given $f_{i^*}$, the transport plan $\sigma$ can be constructed in $O(|E|)$ time since every $s - t$ path in $G$ has constant length and at every iteration the flow along at least one edge is decremented to zero. We conclude with a cost analysis of $\sigma$.

First, we prove the lower bound. By construction of the graph $G$, we note that any path from $s$ to $t$ must be of the form $s \to a \rightsquigarrow b \to t$ where $a \rightsquigarrow b$ is a path in $G$ from $a \in A$ to $b \in B$. By Lemma 2.3, conclude that $\sum_{e \in a \rightsquigarrow b} w_1(e) \geq \mathsf{d}_{G,1}(a,b) \geq \mathsf{d}(a,b)$ for any $a \in A, b \in B$ and any path $a \rightsquigarrow b$ in $G$. Moreover, every edge of the form $a \to a_C$, $b_C \to b$, $s \to a$ and $b \to t$ has cost zero. This implies

$$W_\infty(\mu,\nu) \leq w_\infty(\sigma) = \max_{a,b \in A \times B \,:\, \sigma(a,b)>0} \mathsf{d}(a,b) \leq \max_{e \in E \,:\, f_{i^*}(e)>0} w_1(e).$$

We now prove the upper bound. Suppose we are given the optimal transport plan $\sigma^*$. We construct a flow $f$ with cost at most $(4+\varepsilon) \cdot w_\infty(\sigma^*)$. For each $x,y \in A \times B$, by Lemma 2.3 there exists a path $x \rightsquigarrow y$ in $G$ such that $\sum_{e \in x \rightsquigarrow y} w_1(e) \leq (4+\varepsilon) \cdot \mathsf{d}(x,y)$. Let $\pi(x,y)$ denote this specific path for each pair $x,y$.

Initially, set $f$ to be zero everywhere. Then for each pair $x,y$ where $\sigma^*(x,y) > 0$, we increment flow $f$ on every edge along $s \to x$, $y \to t$ and every edge of $\pi(x,y)$ by $\sigma^*(x,y)$. The resulting flow is a max flow since the capacity on every edge $s \to x$ leaving $s$ is $\mu(x)$ and the capacity on every edge $y \to t$ entering $t$ is $\nu(y)$. Moreover by construction of $f$ and the fact that every edge $s \to x$, $y \to t$, $x \to a_C$ and $b_C \to y$ has cost zero, the cost of the flow $f$ is

$$\max_{e \in E \,:\, f(e)>0} w_1(e) = \max_{x,y \in A \times B \,:\, \sigma^*(x,y)>0} \mathsf{d}_{G,1}(x,y)$$

$$\leq \max_{x,y \in A \times B \,:\, \sigma^*(x,y)>0} (4+\varepsilon) \cdot \mathsf{d}(x,y)$$

$$\leq (4+\varepsilon) \cdot w_\infty(\sigma^*) = (4+\varepsilon) \cdot W_\infty(\mu,\nu).$$

We note that the cost of $f$ is at most $(1+\varepsilon)$ times that of $f_{i^*}$ since $i^*$ is the smallest index where $f_{i^*}$ is a max flow, i.e. has total flow 1, by definition. Rescaling $\varepsilon$ by some constant gives the desired result. □

## F   OMITTED FIGURE FROM SECTION 4

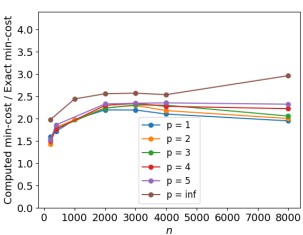

Figure 2: Approximation Ratio (Uniform)

