# OpenReview forum: "A Scalable Constant-Factor Approximation Algorithm for $W_p$ Optimal Transport"
_ICLR.cc/2026/Conference — ICLR 2026 Poster_

### Official Review · Reviewer_HhqV · 2025-10-30

**Soundness:** 2
**Presentation:** 1
**Contribution:** 2
**Rating:** 2
**Confidence:** 3

**Summary:**

This paper studies the problem of computing the $W_p$-distance, for $p \in [1, \infty]$, between two distributions $\mu$ and $\nu$ supported on finite point sets of a metric space. The authors give a $(4 + \epsilon)$-approximate algorithm for computing the optimal transport between $\mu$ and $\nu$ with expected runtime $O(n^2 + (n^{3/2} \epsilon^{-1} \log^2 \Delta \log U)^{1 + o(1)})$, where $\Delta$ is the ratio between the maximum and minimum pairwise distances of the support, and $U$ is the ratio between the maximum and minimum probabilities. Also, when $\mu$ and $\nu$ are uniform distributions supported on two sets of the same size, the authors give a $(4 + \epsilon)$-approximate combinatorial algorithm with expected runtime $O(n^2 \epsilon^{-2} \log^2 \Delta)$ and a $(8 + \epsilon)$-approximate combinatorial algorithm with expected runtime $O(n^2 + n^{5/3} \epsilon^{-2} \log^2 \Delta)$. Finally, the authors show that given an $O(n^2)$-time algorithm for $p = \infty$ that is either $(2 - \epsilon)$-multiplicative approximate or $(\Delta / 2 - \epsilon)$-approximate with constant $\epsilon$, then there exists an $O(n^2)$-time algorithm for computing a perfect matching in general graphs provided exists.

**Strengths:**

This paper studies an interesting and important problem. It is generally well-written, and the proofs are easy to follow. The technical contributions, despite mainly motivated by prior work and some classic techniques, are non-trivial and lead to improvement over prior work in terms of runtime.

**Weaknesses:**

Overall, I find the contributions of this paper marginal. It seems that the main technical contribution comes from the data structure given in Section 2, which at a high level is similar to Bourgain's multi-level sampling. Yet, the analysis of its guarantees follows quite straightforwardly from its construction. The remaining components of the algorithms then apply ideas from some classic algorithms.

Besides, the presentation of this paper requires considerable improvement. A related work section that places this paper into a broader context is missing. The proofs could be restructured following the theorem-lemma style instead of a plain description, and the guarantees of the data structure can be stated more explicitly.

I have some concerns regarding the correctness of the proofs (see Questions).

Detailed comments:
- Line 45-46: I believe that $W_p$ OT being a matching problem also requires $\mu$ and $\nu$ to have the supports of the same size.
- $\Delta$ is used to denote both the aspect ratio and the diameter, which might lead to confusion.
- Line 200: Should $C_y$ be $C_y[i]$? Same for $C_x$.
- Using Algorithm environments to formally describe algorithms presented in the paper would greatly improve the readability.

**Questions:**

- Line 62;64: Why is deciding the existence of a perfect matching in a dense graph is a simpler task?
- Is the approximation ratio $4 + \epsilon$ tight for the given algrorithm?
- Line 166: Would the degree of any particular point be as large as $nt$ instead of $n$?
- Line 178-180: Should the first term in the equation be $\Pr[p \in C_{w_j}[t] \mid w_j \in P_0 \setminus P_1]$? Also, should $j$ in the last term be $j-1$?
- Line 183-185: Should the term in the summation be $\Pr[w_s \in P_0 \setminus P_1] * Pr[p \in C_{w_s}[t] \mid w_s \in P_0 \setminus P_1]$? I believe the range of the second summation should be $s = 0, 1, \ldots, n - 1$. Finally, if the previous question is correct, should the last term be $O(1)$ now that $\Pr[w_s \in P_0 \setminus P_1] = 1 / \sqrt{n}$?
- Line 228-235: This paragraph seems confusing to me. Could you further justify its correctness and runtime? In my understanding here we should do the following: A heap for each $C$ containing $A \cap C$ is maintained, and for each $p$, a heap containing $X =  \{(a_C, C) \mid p \in C\}$ is maintained. So each update $p$ involves updating the heaps associated with $C$ that contains $p$, which might result in updates in the heaps associated with points in $C$. Please correct me if I'm wrong.
- At the end of Section 3.1, Lemma 2.3 is applied to conclude Theorem 1.1 for $p = \infty$. Yet it seems to me that Lemma 2.3 only works for $p < \infty$. Do I miss anything?

---

> ### Author Response · Authors · 2025-11-21
>
> > It seems that the main technical contribution comes from the data structure given in Section 2, which at a high level resembles Bourgain’s multi-level sampling. The analysis appears straightforward from the construction, and the remaining algorithmic components apply classical ideas. It would be helpful to more clearly highlight the technical novelty.
>
> Bourgain’s multi-level sampling has indeed inspired much prior work on clustering for metric spanners and distance oracles. However, our use of this technique in the context of computing the $p$-Wasserstein distance is new. The $p$-Wasserstein setting differs fundamentally from standard spanner constructions: we must approximate $d(⋅,⋅)^p$, which is **not a metric**, and this requires constructing a **directed** spanner that introduces Steiner points and is not strongly connected, yet still preserves $d(⋅,⋅)^p$ along shortest paths. This structural difference leads to a clustering and spanner design tailored specifically to optimal transport.
>
> We also highlight that spanner constructions for general metric spaces are typically quite technical and often require super-quadratic time. In contrast, our approach **substantially simplifies** the construction by introducing Steiner points at cluster centers. These Steiner points allow us to route long-distance edges through a small set of representatives and avoid the dense pairwise checks that make prior constructions expensive.
> Moreover, prior clustering-based distance oracles cannot answer the **weighted bichromatic closest-pair (BCP)** queries essential to our algorithm. To address this, we define a proxy distance on our clusters that supports both weighted nearest-neighbor and weighted BCP queries, completely avoiding pairwise distance computations. This yields a practical, lightweight data structure. The combination of (i) a simplified directed spanner enabled by Steiner points and (ii) proxy distances that support weighted BCP queries constitutes the core technical novelty of our work.
>
> **Minor Clarifications**
>
> > Why is deciding the existence of a perfect matching in a dense graph simpler?
>
> Perfect matching  in bipartite graph is a very special case of optimal transport – assign equal mass to every vertex, weight 1 to every edge in the graph, and a very large weight to all missing edges. From the min-cost flow perspective in a graph, again this is a very special case. No $O(n^2)$-time algorithm is known even for this problem. Of course, one can use Chen et al algorithm to compute a perfect matching.
>
> > Corrections to Lines 178–185 regarding conditional probabilities.
>
> Thank you. This is correct—we will update the line to use the proper conditional probability. For the subsequent step, although we did not write it explicitly,
> $$Pr⁡[w_s \in P_0\setminus P_1]⋅Pr⁡[p \in C_{w_s}[t] \mid w_s \in P_0\setminus P_1]≤Pr⁡[w_s \not\in P_1].$$
> Your observation about the summation range is correct, and we will update this as well.
> Regarding the last term: conditioning on $w_j \in P_1$ means only $w_1,…,w_{j-1}$ matter. The assignments of $w_{j+1},…,w_s$ do not influence whether $p \in C_{w_j}$, so the bound remains valid.
>
> > Lines 228–235 appear confusing; please justify correctness and runtime.
>
> Yes you are correct that we maintain a heap for each cluster ordered by weights. The query procedure checks each cluster that a point belongs to and the update procedure modifies the weight of a point $p$ in all heaps corresponding to clusters containing $p$. Since we have bounded the expected degree of $p$ by $\tilde{O}(n^{1/2})$, this update costs is the desired $\tilde O(n^{1/2})$. We do not need to additionally maintain a separate heap for each point, the global heap is only required in the BCP data structure and we will make this clear in the revised version.
>
> > At the end of Section 3.1, Lemma 2.3 is applied to $p=\infty$, but Lemma 2.3 only applies to $p<\infty$. Why is this valid?
>
> You are correct that Lemma 2.3 applies only to $p<\infty$. For $p=\infty$, the algorithm instead uses the $w_1$ edge costs. Two key observations ensure correctness: (i) the $w_1$ edge costs approximate the metric distances $d$, and (ii) every shortest path from $a$ to $b$ incurs a positive cost on exactly one edge. This allows us to compute approximate $W_\infty$ distances by greedily removing the most expensive edges, computing a max flow, and observing that the $W_\infty$ distance decreases precisely when the max flow has magnitude 1.

---

### Official Review · Reviewer_5TsD · 2025-10-30

**Soundness:** 3
**Presentation:** 3
**Contribution:** 3
**Rating:** 6
**Confidence:** 4

**Summary:**

The paper deals with the $W_p$ optimal transport problem. In this problem we are given two node sets $A$ and $B$, a distance metric $d$ on $A \cup B$ and two probability distributions $\mu$ and $\nu$ defined on $A$ and $B$ respectively. The goal is now to move the probability mass of distribution $\mu$ from $A$ to $B$ such that $\nu$ is formed. Alternatively $\mu$ and $\nu$ could also be seen as node weights on $A$ and $B$ (each summing up to $1$) and we need to match the respective weights to each other. The respective movement or matching should then minimize the $W_p$ distance with respect to the distance metric $d$.

The authors point our that it is already known that the problem can be solved exactly with running time $O\left(n^{2+ o(1)}\right)$ using a min-cost flow algorithm. However, this algorithm is not feasible in practice and there has been significant research over the last years both trying to find practical algorithms as well as trying to obtain a theoretical running time in $O(n^2)$, even if this means that not the optimum solution is found.

The main contribution of the submission is a randomized $2$-layer clustering scheme that allows to create a weighted directed graph of expected size $O(n^{3/2})$ such that the shortest path distance between two nodes $a \in A$ and $b \in B$ in this graph is at least $d(a,b)$ and at most $(4 + \epsilon) d(a,b)$. Crucially all paths from $a$ to $b$ only contain a single weighted edge. As a result also the value $d(a,b)^p$ gets only distorted by a value of $(4+\epsilon)^p$ by this construction (for $p \in \mathbb{N}$).

Afterwards the authors apply the min cost flow algorithm by Chen et al. using these new distances. Given that the cost of the algorithm depends on the number of edges in the respective graph which got reduced from $O(n^{2})$ to $O(n^{3/2})$, this improves the running time accordingly. The distance graph can be calculated in $O(n^2)$, and the author end up with an $(4+ \epsilon)$-approximation with expected running time $O(n^2)$ under reasonable assumptions. For the case that $p = \infty$ they provide an alternative approach with a similar running time.

Besides this, the submission provides an algorithm for the $W_p$ matching problem, which can be seen as the special case of the $W_p$ optimal transport problem where both $\mu$ and $\nu$ are uniform distributions. The advantage of this algorithm seems to be that it has a reasonable practical running time even though the theoretical running time is mostly comparable to the more general algorithm. The approximation ratio stays $(4 +\epsilon)$ (or $(8 + \epsilon)$ respectively). It is also shown that if one finds an $(2 - \epsilon)$-approximation algorithm for the $W_\infty$ matching problem in $O(n^2)$, this would imply that one could find a perfect matching in bipartite graphs in $O(n^2)$. This can be seen as a hardness result.

For their experiments the authors generated data sets using either a uniform or a truncated normal distribution in up to 10 dimensions. They do not evaluate the practical performance of their $W_p$ optimal-transport algorithm directly (probably due to the inefficient min-cost flow algorithm). Instead they only evaluated the quality of the $2$-layer clustering scheme. They showed that the size of the clustering fits the expected theoretical bounds and obtained that in their experiments the maximum distance distortion of this clustering often was between $3$ and $3.5$ while the average distortion was closer to $1.5$. They also provided some experiments for their $W_p$ matching algorithm with the approximation ratios seemingly approaching values between $2$ and $2.5$ with larger number of nodes (at most 8000). An exception seems to be the algorithm for $p = \infty$. Here the ratio reaches the value of $3$ and it is unclear if the value would increase even more for larger point sets. The algorithm also seems to be reasonably efficient.

**Strengths:**

The paper contains non-trivial theoretical results. At first glance the improvement from an exact $O(n^{2+ o(1)})$ algorithm to an $O(n^2)$-approximation algorithm (under certain assumptions) seems to be not that impressive. However, the authors provide a lot of literature that deal with this problem and their algorithm improves the best existing result significantly.

**Weaknesses:**

The fact that the new $W_p$ optimal-transport algorithm was not tested in the experiments seems to indicate that it is very inefficient in practice. It is unfortunate that the authors did not provide experimental results for existing algorithms for a better comparison.

**Questions:**

- The data structures for the proximity queries are rather involved and are not directly necessary for the optimal-transport algorithm. Maybe one could improve the readability of the paper by directly providing the algorithm after the introduction of the layer clustering and present the query structures afterwards. This would also be beneficial because the need to answer these queries only gets clear once the matching algorithm gets presented. Thus presenting this algorithm directly afterwards could be helpful.

- Given that the paper focuses a lot on improving the running time it could make sense to provide a summary of the analysis of the running time of the $W_p$ optimal transport algorithm somewhere in the paper or the Appendix (which could also discusses the running time for computing the clustering).

- In the experimental section there is no comparison with existing algorithms which would have been very helpful to judge whether the new results also yield improvements in practice.

- In Theorem 1.3 the authors say that one could calculate the perfect matching for an arbitrary graphs (if it exists) but in the appendix they only prove that one could find the perfect matching in a bipartite graph.

- In line 471 the authors write that the $W_p$ matching algorithm performs 'near optimal' in practice. This seems to be a bit of a stretch for approximation ratios around $2$.

---

> ### Author Response · Authors · 2025-11-21
>
> > The fact that the new  optimal-transport algorithm was not tested in the experiments seems to indicate that it is very inefficient in practice. It is unfortunate that the authors did not provide experimental results for existing algorithms for a better comparison.
>
> We have provided a code for the special case of assignment problem and we plan to include a comparison with the algorithm by Lahn et al. in the next version.
>
> > The data structures for the proximity queries are rather involved and are not directly necessary for the optimal-transport algorithm. Maybe one could improve the readability of the paper by directly providing the algorithm after the introduction of the layer clustering and present the query structures afterwards. This would also be beneficial because the need to answer these queries only gets clear once the matching algorithm gets presented. Thus presenting this algorithm directly afterwards could be helpful.
>
> This is a good point. We had thought about this and decided to use the current structure for the following reason. The BCP data structure relies closely on the clustering method, and its definition naturally builds on the properties established during clustering. Conversely, the matching algorithm treats the data structure as a black box. If we were to present the algorithm immediately after clustering, this would separate the clustering from the construction of the data structure on which the algorithm depends, potentially making the flow more confusing for readers.
>
> > Given that the paper focuses a lot on improving the running time it could make sense to provide a summary of the analysis of the running time of the  optimal transport algorithm somewhere in the paper or the Appendix (which could also discuss the running time for computing the clustering).
>
> We will add a summary of our analysis in the next version.
>
> > In the experimental section there is no comparison with existing algorithms which would have been very helpful to judge whether the new results also yield improvements in practice.
>
> For $p=\infty$, there are no known implementable quadratic-time approximation algorithms that we can compare against. The result of Lahn et al. (ICML 2025) relies on the construction of $p$ distinct HSTs and therefore does not apply when $p=\infty$. For fixed values of $p$, although from a theoretical standpoint the algorithm of Lahn et al. provides an $O(\log ⁡n)$-approximation and ours provides an $O(1)$-approximation, their empirical performances are comparable to ours on some of the data sets we tested. We will include a comparison of our approach and the approach of Lahn et al. in the next version of the paper.
>
> > In Theorem 1.3 the authors say that one could calculate the perfect matching for an arbitrary graphs (if it exists) but in the appendix they only prove that one could find the perfect matching in a bipartite graph.
>
> We mean arbitrary **bipartite** graphs, and we will clarify this explicitly in the revised version.
>
> > In line 471 the authors write that the matching algorithm performs 'near optimal' in practice. This seems to be a bit of a stretch for approximation ratios around.
>
> We intended to say that the algorithm performs close to the **worst-case lower bound** that we establish theoretically. We acknowledge that the data used in our experiments is not worst-case, and therefore the phrasing “near-optimal” is not appropriate. We will update this wording in the next version.

---

> > ### Comment · Reviewer_5TsD · 2025-11-23
> >
> > Dear Authors,
> >
> > Thank you very much for your detailed answer and the additional information provided. While some details are now clearer to me, my overall impression has not changed and I will keep my current score.
> >
> > Best wishes,
> > Reviewer

---

### Official Review · Reviewer_okv2 · 2025-11-02

**Soundness:** 4
**Presentation:** 3
**Contribution:** 3
**Rating:** 6
**Confidence:** 4

**Summary:**

This paper studies the problem of computing the $p$-Wasserstein distance in general metric spaces for any $p\in[1,+\infty]$. The problem can be formulated as a minimum-cost flow problem and therefore can be solved exactly in $n^{2+o(1)}$ time using the sophisticated and impractical theoretical algorithm of Chen et al. (FOCS 2022). This paper considers approximation algorithms for the Wasserstein distance and obtains near-quadratic $\tilde{O}(n^2)$-time algorithms with $4+\epsilon$ approximation ratio, where $n$ is the support size of the two input distributions. This improves upon the previous $O(\log n)$-approximation in $\tilde{O}(n^2)$ time (Lahn et al., ICML 2025). The authors also present a simpler and more practical algorithm for the special case where the input distributions are uniform (i.e., the $W_p$ matching problem), as well as a nearly matching conditional lower bound.

Technically, both algorithms build on a clustering scheme obtained via a multi-level sampling procedure inspired by Bourgain (1985). The authors use this scheme to

- construct a directed spanner with $m = \tilde{O}(n^{3/2})$ edges that preserves the $p$-power metric $d(\cdot,\cdot)^p$, and by combining it with the $m^{1+o(1)}$-time algorithm for graphs (Chen et al., FOCS 2022), obtain a near-quadratic $\tilde{O}(n^2)$-time algorithm for $p$-Wasserstein distance; and

- design data structures for weighted nearest-neighbor search and dynamic bichromatic closest-pair queries, which are then used to speed up the Gabow–Tarjan bipartite matching algorithm, leading to a near-quadratic $\tilde{O}(n^2)$-time algorithm for $W_p$ matching.

**Strengths:**

- The paper addresses the efficiency challenge of the fundamental $p$-Wasserstein distance problem and achieves significant improvements (reducing the approximation ratio from $O(\log n)$ to  $O(1)$). Moreover, their algorithms are general and work for any $p \in [1,+\infty]$, whereas the previous $O(\log n)$-approximation only works for small $p$.
- The algorithms are also practical, as demonstrated by the experimental results.
- The results are also complete, as the authors additionally provide a (conditional) lower bound.
- The paper is well-written and I can easily follow the presentation and understand the main idea of the algorithm and its analysis.

**Weaknesses:**

The only weakness in my view is that the technical contribution is not clearly explained. The paper’s main technique, a clustering scheme via multi-level sampling, seems largely based on prior work (e.g., Bourgain, 1985). The directed spanner construction and the data structures appear to follow naturally from this clustering scheme. It would be helpful if the authors could more explicitly highlight their technical novelty (for example, by clarifying how their multi-level sampling differs from prior work).

In addition, I believe the experimental section could be further improved. In particular, it would be useful to compare against the previous $O(\log n)$-approximation algorithm of Lahn et al. (ICML 2025), and to evaluate the algorithm on some real-world datasets.

**Questions:**

In the “Algorithm efficiency” paragraph of the experiment section, the authors report the number of operations but not the actual running time. Could the authors clarify the motivation for this choice?

Moreover, in Line 465 the paper states that *“Combined with the $O(n^2)$ per-query complexity … the algorithm runs in quadratic time ...”*. However, the results seem to indicate that there are at least $n^{3/2}$ queries, which would imply a total running time of $O(n^{3.5})$ rather than $O(n^2)$. Is this a typo, or am I misunderstanding something?

---

> ### Author Response · Authors · 2025-11-21
>
> > The only weakness in my view is that the technical contribution is not clearly explained. The paper’s main technique, a clustering scheme via multi-level sampling, seems largely based on prior work (e.g., Bourgain, 1985). The directed spanner construction and the data structures appear to follow naturally from this clustering scheme. It would be helpful if the authors could more explicitly highlight their technical novelty (for example, by clarifying how their multi-level sampling differs from prior work).
>
> Bourgain’s multi-level sampling has indeed motivated much prior work on clustering for metric spanners and distance oracles. However, our use of this technique in the context of computing the $p$-Wasserstein distance is new. The $p$-Wasserstein setting differs fundamentally from standard spanner constructions: we must approximate $d(x,y)^p$, which is not a metric, and this requires constructing a **directed spanner** that consists of Steiner points and that is not strongly connected but still preserves $d(⋅,⋅)^p$ along shortest paths. This structural difference leads to a clustering and spanner design that is tailored specifically to OT.
> We also want to highlight that **existing spanner constructions for general metric spaces are technically involved** and typically require **super-quadratic time**. In contrast, our approach substantially **simplifies the construction** by introducing **Steiner points** at cluster centers. These Steiner points allow us to route long-distance edges through a small set of representative nodes, avoiding the dense pairwise checking that makes prior constructions expensive. As a result, we obtain a **quadratic-time spanner construction** even for arbitrary metric spaces—a significant simplification over existing techniques.
>
> Moreover, existing clustering-based distance oracles cannot answer the **weighted bichromatic closest-pair (BCP)** queries essential to our algorithm. To handle this, we define a **proxy distance** on clusters that supports both weighted nearest-neighbor and weighted BCP queries. This avoids expensive pairwise distance computations entirely and yields a practical, lightweight data structure. The combination of (i) a simplified directed spanner enabled by Steiner points and (ii) proxy distances supporting weighted BCP queries constitutes the core technical novelty of our work.
>
> > In addition, I believe the experimental section could be further improved. In particular, it would be useful to compare against the previous -approximation algorithm of Lahn et al. (ICML 2025), and to evaluate the algorithm on some real-world datasets.
>
> We are including a comparison with Lahn et al. and are also testing our algorithm on matching two sets of MNIST images in the updated experimental section.
>
> > In the “Algorithm efficiency” paragraph of the experiment section, the authors report the number of operations but not the actual running time. Could the authors clarify the motivation for this choice?
>
> Our code is written in Python, which is not well suited for high-performance graph algorithms. A C or C++ implementation would be necessary to obtain fast absolute running times. Our goal in the current experiments was to empirically demonstrate that the number of operations matches the theoretical predictions, rather than to optimize low-level runtime.
>
> > Moreover, in Line 465 the paper states that “Combined with the  per-query complexity … the algorithm runs in quadratic time ...”. However, the results seem to indicate that there are at least  queries, which would imply a total running time of  rather than . Is this a typo, or am I misunderstanding something?
>
> This is a typo. The per-query complexity is $\sqrt{n}$, and we will correct this in the revised version

---

### Official Review · Reviewer_6YEK · 2025-11-03

**Soundness:** 2
**Presentation:** 3
**Contribution:** 3
**Rating:** 4
**Confidence:** 4

**Summary:**

The paper presents two constant factor approximation algorithms for optimal transport. The first one, relies on computing a min-cost flow, but does so on a smaller graph that approximates distances of the full graph. The second algorithm is for the restricted setting where the distributions are uniform.

**Strengths:**

Optimal transport is a heavily studied and new algorithmic insights should always be welcome.

**Weaknesses:**

The paper emphasizes the "scalable" nature of the algorithm, but the general algorithm still relies on Chen et al. for computing a min-cost flow. Moreover, I believe some more detail is needed to justify the expected running time. A graph is constructed, which has m edges in *expectation* and then the algorithm by Chen et al. is applied to that graph. However, as far as I can see, this does not immediately result in an algorithm that runs in time $m^{1+o(1)}$. To conclude this, some concentration bound on the number of edges or some suitable strict upper bound on the number of edges would have to be used. In any case, this would require a little more justification.

For dense graphs, there are also other nearly linear time algorithms available (van den Brand et al. STOC 2021). Maybe the current submission can claim to save polylogarithmic factors for maximally dense graphs? But the paper presents itself as aiming to be more practical and a result like that would not necessarily qualify for a practical improvement.

Not surprisingly, no experiments are included for the general optimal transport algorithm.

I am not claiming that the theoretical contribution is without merit (after adding an appropriate justification for the running time bound), but that there is a gap between the claimed practical focus and this result.

The algorithm for uniform distributions does not rely on Chen et al. and so the critique above does not apply.

**Questions:**

In line with the above, could please give a more detailed justification on the claimed running time?

Do you have experiments comparing your matching algorithms to others both in terms of efficiency and accuracy?

---

> ### Author Response · Authors · 2025-11-21
>
> > The paper emphasizes the "scalable" nature of the algorithm, but the general algorithm still relies on Chen et al. for computing a min-cost flow.
>
> We agree that the near-quadratic algorithm described in the paper uses the min-cost flow algorithm of Chen et al. as a black-box subroutine. However, the clustering scheme described in the paper can be combined with a fully combinatorial capacity-scaling framework to obtain a slightly super-quadratic, but still scalable, constant-factor approximation algorithm that does not rely on Chen et al. In particular, one can compute optimal transport using $O(n^2\log⁡ U)$ queries to a weighted bichromatic closest-pair (BCP) data structure, where $\log U$ is the number of bits needed to represent the probability masses. Our clustering scheme can be used to answer dynamic $O(1/\varepsilon)$-approximate BCP queries in $O(n^\varepsilon)$ time, leading to an overall execution time of $O(n^{2+\varepsilon}\log⁡ U)$.
> Thus, our clustering-based framework actually yields two scalable approximation schemes: (i) the main nearly quadratic algorithm that leverages Chen et al. for min-cost flow, and (ii) a slightly super-quadratic but fully combinatorial $O(1/\varepsilon)$-approximation algorithm based on capacity scaling and approximate BCP queries. This shows that our “scalable’’ claim does not hinge on a single specific flow implementation, but on a structural reduction that can be instantiated in multiple algorithmic regimes. **We will include this additional result in our updated version of the paper**.
>
> > However, as far as I can see, this does not immediately result in an algorithm that runs in time. To conclude this, some concentration bound on the number of edges.
>
> This can be handled using a standard trick. We repeatedly apply the clustering procedure until we obtain a spanner of size $n^{3/2}$. By Markov’s inequality, the probability that the number of edges exceeds twice the expected value is at most $1/2$. Therefore, the expected number of repetitions needed to obtain a spanner of size $O(n^{3/2})$ is at most $2$. As a result, we maintain the same expected execution time while guaranteeing that we can apply the algorithm of Chen et al. on a graph of size $O(n^{3/2})$.
>
> > I am not claiming that the theoretical contribution is without merit ..., but that there is a gap between the claimed practical focus and this result.
>
> We appreciate the reviewer’s comment regarding the practical focus of the paper. We would like to clarify that our algorithm is not intended solely as a *single-shot* OT solver. A key property—implicit in the submitted version but not emphasized—is that the method naturally supports the query model, where the support set
> $X=\{x_1,\dots,x_n\}\subset M$
> is fixed in advance and one must repeatedly compute (approximate) $p$-Wasserstein distances between different distributions supported on X. This regime is common in many practical settings. For example, in NLP, pre-trained word embeddings map each vocabulary word to a point in $R^d$, and each document (e.g., in Word Mover’s Distance) becomes a distribution over this fixed embedding set. Similar scenarios where we have different distributions with the same support arise in computer vision (histograms over learned feature dictionaries), shape analysis (shared point geometry with different mass distributions), and generative modeling (distributions over a shared latent codebook).
>
> In precisely these scenarios, our approach incurs a **one-time** $O(n^2)$ preprocessing cost to construct the hierarchical clustering and directed spanner on the support. After this, **each individual OT computation runs in the near-linear/sub-quadratic time** stated in Theorem 1.1. Thus, while a single invocation may not appear faster than classical solvers, the amortized per-query cost is dramatically smaller whenever multiple OT computations share the same support—an extremely common practical regime. This substantially narrows the perceived gap between the theoretical result and the paper’s practical motivation.
> Although this query-model interpretation is a direct and natural consequence of our framework, it was not made explicit in the submission. We agree to highlight this feature of our algorithm.
>
> > Do you have experiments comparing your matching algorithms to others both in terms of efficiency and accuracy?
>
> For $p=\infty$, there are no known implementable quadratic-time approximation algorithms that we can compare against. The result of Lahn et al. (ICML 2025) relies on the construction of $p$ distinct HSTs and therefore does not apply when $p=\infty$. For fixed values of $p$, although from a theoretical standpoint the algorithm of Lahn et al. provides an $O(\log ⁡n)$-approximation and ours provides an $O(1)$-approximation, their empirical performances are comparable to ours on some of the data sets we tested. We will include a comparison of our approach and the approach of Lahn et al. in the next version of the paper.

---

### Official Review · Reviewer_coG2 · 2025-11-10

**Soundness:** 2
**Presentation:** 2
**Contribution:** 2
**Rating:** 6
**Confidence:** 2

**Summary:**

The paper presents the first scalable, constant-factor approximation algorithm for the $W_{p}$ optimal transport (OT) problem that runs in nearly quadratic time. This method significantly improves upon previous $O(\log n)$-approximation algorithms and is the first efficient, quadratic-time method to extend to the $W_{\infty}$ distance.

** Problem formulation.**

The paper addresses the $W_{p}$ optimal transport problem for discrete probability distributions $\mu$ and $\nu$, supported on finite point sets $A$ and $B$ within a metric space $(X,d)$. The objective is to find a transport plan $\sigma: A \times B \rightarrow R_{\geq 0}$ that minimizes the $W_{p}$ cost, defined as $w_{p}(\sigma):=(\sum_{a\in A,b\in B}\sigma(a,b)\times d(a,b)^{p})^{1/p}$. This formulation also covers the $W_{\infty}$ cost, which is the limit as $p \rightarrow \infty$ and represents the maximum distance $d(a,b)$ for which $\sigma(a,b) > 0$.

** Main results **

The main result (Theorem 1.1) is a randomized algorithm that computes a $(4+\epsilon)$-approximate $W_{p}$ optimal transport plan in $O(n^{2}+(n^{3/2}\epsilon^{-1}\log^{2}\Delta~\log U)^{1+o(1)})$ expected time for any $p \in [1, \infty]$. Additionally, a simpler $\tilde{O}(n^{2})$ combinatorial algorithm is provided for the $W_p$ matching problem (when distributions are uniform).

** Technique/algorithm **

The core technique is a two-layered clustering scheme inspired by Bourgain's multi-level sampling, which approximates the cost function $d(\cdot,\cdot)^p$. The primary algorithm constructs a directed spanner graph based on these clusters and computes an approximate solution by running a minimum-cost max-flow algorithm on this graph. A second, simpler algorithm for the matching problem uses the same clustering to build efficient dynamic data structures (for bichromatic closest pair and weighted nearest neighbor queries) which are then used within a Gabow-Tarjan cost-scaling matching framework.

** Experiment sumamry **

The paper provides an empirical evaluation of the simpler combinatorial matching algorithm (from Section 3.2) on synthetic data. These experiments demonstrate that the algorithm's practical approximation ratio is consistently much better than the theoretical $(4+\epsilon)$ worst-case bound (typically around 1.5-2.0) and that the runtime scales quadratically, as predicted by the analysis.

**Strengths:**

The theoretical results are solid and also discusses its potential practical application

**Weaknesses:**

.

**Questions:**

.

**Details Of Ethics Concerns:**

.

---

### Author Response · Authors · 2025-11-21
**Common Response to All Reviewers**

We thank the reviewers for their valuable feedback and are preparing a new version of the paper that incorporates this feedback, which we will upload soon.
Across the reviews, several comments touched on the practical significance of our contribution. We would like to emphasize three aspects that clarify why our results are highly relevant in practice.
**First**, although this was *not highlighted in the submitted version*, our algorithm naturally operates in the **query model**, which is an immediate consequence of our clustering-based framework. In many practical scenarios—such as OT over word embeddings in NLP, histograms over learned visual dictionaries, or distributions over latent codebooks in generative models—the support set is fixed while many different distributions must be compared. In this regime, our method performs a one-time $O(n^2)$ preprocessing step to build the clustering and spanner structure, after which *each* subsequent OT computation runs in $O(n^{1+\varepsilon})$ time for a $O(1/\varepsilon)$-approximation. We will make this query-model interpretation explicit in the revised version, as it substantially strengthens the practical relevance of our algorithm.

**Second**, although this was *not mentioned in the submitted version*, our clustering framework also yields a **fully combinatorial** $O(1/\varepsilon)$**-approximation algorithm** with running time $O(n^{2+\varepsilon}\log⁡ U)$ when combined with a standard capacity-scaling approach. Thus, the scalability of our method is not tied to the min-cost flow algorithm of Chen et al.; the framework can be instantiated in multiple algorithmic regimes and remains robust even without Chen et al. We will include this result in the updated version of the paper.

**Third**, to the best of our knowledge, we provide the **first practical quadratic-time approximation algorithm for $W_\infty$** that works for **arbitrary metric spaces**. Existing approaches either require special metric structure (e.g., grids, low-dimensional Euclidean spaces) or rely on constructions (such as $p$HSTs) that fundamentally do not apply when $p=\infty$. Our work fills this gap by giving the first general, implementable algorithm with provable guarantees.

Taken together, these three points show that our contribution is not only theoretically novel but also practically meaningful: it yields real computational benefits in common application regimes, supports multiple scalable implementations, and provides the first general-purpose approximation algorithm for $W_\infty$ across arbitrary metric spaces.

---

### Author Response · Authors · 2025-11-27
**Statement for Rebuttal Revision**

We have substantially updated the paper to address the reviewers’ feedback and to strengthen both the theoretical and empirical contributions. First, we have revised the results section to explicitly incorporate and formalize the **query-model version** of our algorithm, highlighting its practical relevance for settings that require repeated OT computations. Second, we now **state and integrate** our new **combinatorial optimal transport result**, which follows naturally from our clustering–BCP/WNN framework. Third, we have **clarified and expanded our explanation of the technical contributions**, including a more detailed discussion of the clustering scheme, the directed spanner construction, and the role of Steiner points. Finally, we have expanded the experimental section to (i) include a direct comparison with the algorithm of Lahn et al.\ (ICML 2025), and (ii) evaluate our method on high-dimensional real data from the **MNIST dataset**, in addition to our existing synthetic benchmarks. **All changes made in response to the reviewers’ comments have been marked in blue in the updated manuscript.** These updates together substantially strengthen the clarity, completeness, and practical impact of our contributions.

---

### Meta-Review · Area_Chair_QAXt · 2026-01-07

**Summary:**

Concerns about practicality and scalability: The general OT algorithm relies on Chen et al.’s min-cost flow as a black box, which raised doubts about whether the strong “scalable” and “practical” claims are justified. In addition, no experimental evaluation is provided for the general OT algorithm.

Insufficient justification of the claimed running time: Reviewers questioned whether the expected spanner size and overall runtime bounds follow directly, and asked for clearer arguments

Unclear technical novelty: The clustering approach appears closely related to Bourgain-style multi-level sampling, and reviewers found that the paper does not clearly state what is technically new beyond prior work.

Presentation and organization issues: Reviewers noted the lack of a well-developed related work discussion and asked for clearer, more structured presentation of algorithms and theoretical results.

Technical correctness and clarity at a detailed level: Several reviewers raised specific questions about lemma applicability, complexity analysis, notation, and possible typos or ambiguities.

Experimental reporting concerns: The choice to report operation counts instead of actual running time, as well as inconsistencies or typos in complexity statements, was viewed as problematic.

**Reviewer Concerns:**

Practicality and scalability: The revised paper clearly formalizes the query-model setting and incorporates fully combinatorial approximation algorithms that do not rely on Chen et al., which substantially strengthens the scalability claims. However, there is still no experimental evaluation of the general OT algorithm based on min-cost flow.

Justification of running time: The revised manuscript explicitly analyzes the expected cluster degree, spanner size, and resulting runtime, providing a coherent justification of the claimed expected bounds. This concern is largely resolved.

Technical novelty: The revision clarifies the novelty by emphasizing OT-specific challenges, directed spanners with Steiner points, and support for weighted BCP/WNN queries.

Presentation and organization: The paper improves clarity by expanding explanations, adding a query-model discussion, and better contextualizing contributions. Nonetheless, some presentation concerns are only partially addressed.

Technical correctness and clarity: Most reviewer-raised technical issues directly addressed and corrected in the revised version.

Experimental evaluation: The experimental section is significantly strengthened with comparisons to Lahn et al. and evaluations on MNIST. Still, experiments focus on the matching/combinatorial setting, and validation of the general OT algorithm remains missing.

**Reviewer Scores:**

Overall, the rebuttal and revised manuscript address most of the concrete technical and clarification-related concerns raised in the reviews. Reviewers who were initially positive or borderline positive would likely maintain their scores with increased confidence due to the clearer articulation of novelty, strengthened runtime analysis, and expanded experimental evaluation.
A borderline negative reviewer would plausibly shift toward a more favorable assessment, as the main objections regarding scalability and complexity justification were substantially addressed.
Although many specific issues raised by the most critical reviewer were improved, their fundamental skepticism about the paper’s novelty and significance is judgment-based and therefore less likely to be fully resolved.

---

### Decision · Program_Chairs · 2026-01-26

Accept (Poster)